# Hard-Constrained Graph Generation with Discrete-Projection Diffusion

**Xuesong Zhang** [1]  **Haifeng Sun** [1]  **Qi Qi** [1]  **Shengkuan Li** [1]  **Yuhao Li** [1]  **Tianyi Kou** [1]  **Zirui Zhuang** [1]  **Bo He** [1]
**Jianxin Liao** [1]  **Jingyu Wang** [1]

## Abstract

Diffusion models have achieved remarkable success in graph generation, but enforcing hard constraints on generated graphs remains challenging, limiting their deployment in constraint-critical applications. Existing approaches either fail to guarantee strict constraint satisfaction or are limited to narrow constraint types, lacking the flexibility to handle diverse constraint specifications. To address this challenge, we exploit the discrete structure of graphs, which allows hard constraints to be formulated as symbolic reasoning problems. Building on this insight, we propose NSPSG, a framework that integrates unconstrained diffusion models with discrete projection operators. NSPSG employs an SMT (Satisfiability Modulo Theories)-based projector to ensure that the generated graphs strictly satisfy constraints while remaining within the training data distribution. To further accelerate generation, we employ a supervised auto-regressive neural projector to approximate the symbolic reasoning process. Across heterogeneous constraints and various graph generation datasets, NSPSG achieves 99%-100% validity rates, demonstrating state-of-the-art performance. Notably, for a complex non-linear constraint, it improves data validity by up to 43% and reaches 99% validity while maintaining comparable generation. Code is available at https://github.com/zhangxuesong2000/Neuro-Symbolic-Projected-Sampling-for-Graphs.

## 1. Introduction

Graph generation seeks to learn the probability distribution of graph-structured data and draw new samples from it, supporting a wide range of applications such as molecular design (Shen et al., 2024; Huang et al., 2023) and social network analysis (Wang et al., 2021; Liu et al., 2025). Deep generative models serve as effective tools for learning such distributions (Vignac et al., 2022; Chen et al., 2025b; Xu et al., 2024), enabling the generation of realistic and diverse graphs that can accelerate scientific discovery and preserve privacy in sensitive domains (Bergmeister et al., 2023; Chen et al., 2025a; Huang et al., 2024). Among these approaches, diffusion models have recently emerged as particularly effective (Mercatali et al., 2024; Jo et al., 2022), achieving superior performance through progressive denoising.

However, most diffusion-based graph generators cannot guarantee that generated graphs strictly satisfy domain-specific hard constraints, limiting their further applicability (Zhao et al., 2024). These constraints are essential for semantic validity and practical utility. For example, neural architecture search requires dimensional consistency in computational graphs for valid execution (Li et al., 2024; Asthana et al., 2024), and molecular graph design demands strict valence rules to ensure chemical stability (Huang et al., 2024). Modern unconditional diffusion models, even when trained on constraint-satisfying datasets, lack such guarantees for generated samples due to the inherently stochastic nature of the reverse process (Christopher et al., 2024; Liang et al., 2025; Madeira et al., 2024). In practice, when generating digital pathology graphs (Madeira et al., 2024), the DiGress model (Vignac et al., 2022) exhibits violation rates as high as 87%, requiring nearly $8\times$ the generation cost to obtain valid samples. Moreover, even on graphs with basic structural constraints, which to our knowledge yield relatively high validity rates, it cannot exceed 90% validity, hindering reliable deployment.

Recent works attempt to enforce constraints by integrating rule-based corrections into the sampling procedure, relying on handcrafted forward processes and acceptance rules designed for specific constraint types (Sharma et al., 2024; Madeira et al., 2024; Chen et al., 2023). Construct (Madeira et al., 2024) exemplifies this by using an edge-absorbing diffusion model and incrementally adding edges during reverse sampling while rejecting violations of properties such as planarity. Although effective, these constraint-specific methods require ad-hoc redesigns for new constraints and struggle to handle diverse or multi-constraint scenarios.

---

[1]State Key Laboratory of Networking and Switching Technology, Beijing University of Posts and Telecommunications, Beijing, China. Correspondence to: Haifeng Sun <hfsun@bupt.edu.cn>.

*Proceedings of the 43rd International Conference on Machine Learning*, Seoul, South Korea. PMLR 306, 2026. Copyright 2026 by the author(s).

To address this problem, we exploit the discrete structure of graphs. Concretely, hard constraints are conditions on the graph structure, such as edge existence or degree restrictions, which can be directly modified. This discrete property naturally enables a formulation of constraint satisfaction as a symbolic reasoning problem, where constraints are expressed as logical specifications, and violations can be precisely identified and corrected through direct modifications. We therefore employ a discrete-projection diffusion process (Cardei et al., 2025; Liang et al., 2025; Sharma et al., 2024), mapping the unconstrained samples from the diffusion model onto the feasible space through symbolic reasoning. This decouples distribution learning from constraint enforcement, enabling a single diffusion model to handle various constraint scenarios.

We realize this discrete projection operator through Satisfiability Modulo Theories (SMT) solvers (Barbosa et al., 2022; Scott et al., 2023; Xu et al., 2021; Albaba & Ozer, 2021). As a symbolic reasoning approach, SMT solvers can encode various hard constraints as logical and arithmetic formulas and solve graph structures that satisfy all specified constraints (Dutertre & De Moura, 2006; Cimatti et al., 2013; Reynolds & Kuncak, 2015). To further preserve the fidelity to the learned distribution during projection, we employ Maximum Satisfiability Modulo Theories (MaxSMT) (Martins et al., 2014), an extension of SMT that minimizes modifications to the original diffusion output while still satisfying hard constraints.

In this paper, we propose NSPSG (Neuro-Symbolic Projected Sampling for Graphs), a hybrid approach that combines symbolic reasoning with neural approximation for efficient constrained graph generation. To reduce the cost of repeatedly calling solvers, we further introduce an autoregressive neural projector that learns to approximate projection mapping (Wang et al., 2024). During training, the SMT solver projects constraint-violating samples to feasible solutions, creating input-output pairs for supervised learning (Diamant et al., 2023; Yolcu & Póczos, 2019). At inference time, the trained projector performs projection in a single forward pass, reducing computational cost while preserving satisfaction rates comparable to the SMT solver.

Comprehensive experiments demonstrate the effectiveness of our method. On single-constraint benchmarks, NSPSG achieves 100% validity for both arithmetic (Sharma et al., 2024) and structural constraints (Madeira et al., 2024), improving over baselines by 24% and 10%, respectively, with comparable computational cost. On a dataset with complex non-linear constraints (Li et al., 2024), NSPSG reaches 99% validity with 43% improvements, while incurring only a modest increase in computational time (less than 10%).

In summary, our contributions are as follows:

1) Discrete Projection for Exact Constraint Enforcement. To the best of our knowledge, we are the first to decouple constrained graph generation into unconstrained sampling and discrete projection that operates directly on graph structures, ensuring exact constraint satisfaction.

2) Neuro-Symbolic Projection Framework. We propose NSPSG, which combines SMT-based symbolic reasoning with neural projection learning. This framework achieves flexibility in handling diverse constraints and enables plug-and-play constraint control.

3) Comprehensive Validation. Through extensive experiments on various benchmarks, we demonstrate that NSPSG achieves near-perfect constraint satisfaction while outperforming baselines in generation quality and maintaining high computational efficiency.

## 2. Preliminaries

### 2.1. Problem Setup and Notations

The constrained graph generation aims to generate graphs that satisfy specified constraints. Formally, given an unconditional diffusion model with an induced distribution $\hat{p}$ and the space of feasible graphs $\mathcal{C}$, the task is to sample a set of graphs $\{G\}$ from $\hat{p}$ that satisfy $G \in \mathcal{C}$ (Qin et al., 2024).

Following standard practice in discrete graph generation (Vignac et al., 2022; Xu et al., 2024; Liu et al., 2024), a graph is represented as $G = (X, E)$, where $X$ and $E$ denote the sets of nodes and edges. For unweighted graphs, $e_{ij} \in \{0, 1\}$ and node attributes are categorical $x_i \in \{1, \ldots, a\}$, where $a$ is the cardinality attribute.

Constraints are formalized as binary predicates $c_k : \mathcal{G} \to \{0, 1\}$ for $k = 1, \ldots, K$, where $\mathcal{G}$ denotes all possible graphs and $c_k(G) = 1$ indicates that the graph $G$ satisfies the $k$-th constraint (Sharma et al., 2024). The feasible region is then defined as $\mathcal{C} = \{ G \in \mathcal{G} \mid c_k(G) = 1, \forall k = 1, \ldots, K \}$. As a concrete example, impose a simple degree upper bound 5 by defining $c(G) = 1$ if $\forall v \in X$, Degree$(v) \le 5$ and 0 otherwise.

### 2.2. Discrete Diffusion and Its Limitations

Discrete diffusion models have demonstrated strong empirical performance for graph generation (Xu et al., 2024; Siraudin et al., 2024). These models operate by corrupting clean graphs into noise through a forward process, then learning to reverse this corruption for generation.

**Discrete Diffusion Generation Process.** We first review how discrete diffusion generates graphs. Given a training graph $G_0 = (X_0, E_0)$, the forward process progressively adds noise via a Markov chain, producing a sequence of noisy graphs $G_1, \ldots, G_T$. The transition prob-

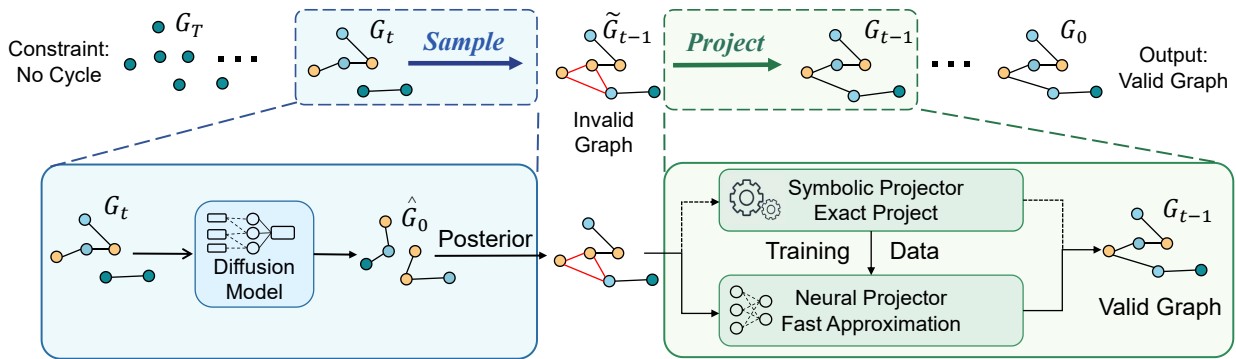

*Figure 1.* Constrained graph generation process in NSPSG. Given current sample $G_t$, the diffusion model samples $\tilde{G}_{t-1}$ from the posterior distribution $q(G_{t-1} \mid G_t, \hat{G}_0)$, which may violate constraints. A projector then maps $\tilde{G}_{t-1}$ onto the feasible set to obtain the valid graph $G_{t-1}$. To balance validity and efficiency, the symbolic projector generates training data with exact constraint satisfaction, which the neural projector learns to approximate for accelerated sampling.

ability is defined using categorical matrices $Q_X^t$ and $Q_E^t$ as $q(G_t|G_{t-1}) = (X_{t-1}Q_X^t, E_{t-1}Q_E^t)$, which randomly flips node types and edge connections. As $t \to T$, the graph loses its original structure and converges to a stationary noise distribution $G_T$.

The reverse process generates new graphs by learning to denoise. A neural network $\mathcal{M}$ is trained to predict the clean graph $\hat{G}_0$ from any noisy state $G_t$. This prediction determines the reverse transition probabilities $p(G_{t-1} \mid G_t) \approx q(G_{t-1} \mid G_t, \hat{G}_0)$ (Vignac et al., 2022; Chen et al., 2023). During sampling, the model starts from pure noise $G_T$ and iteratively denoises by sampling $\tilde{G}_{t-1} \sim p(G_{t-1} \mid G_t)$, gradually recovering a coherent graph $\tilde{G}_0$.

**Limitations for Constrained Generation.** This probabilistic sampling mechanism fundamentally conflicts with hard constraint enforcement. Existing models use stochastic sampling (e.g., Categorical or Bernoulli distributions) (Vignac et al., 2022; Tseng et al., 2023), which inherently assigns non-zero probabilities to constraint-violating samples. Consequently, each denoising step risks generating infeasible samples, making strict validity guarantees difficult to achieve. This limitation is particularly severe for global constraints (e.g., connectivity or degree bounds), which require coordinated decisions across the entire graph that local stochastic sampling cannot adequately capture.

### 2.3. Projection-Based Constrained Generation

Projection-based sampling enforces hard constraints by projecting generated samples onto the feasible region (Christopher et al., 2024; Liang et al., 2025). At each denoising step, it first samples $\tilde{x}_{t-1}$ and then applies the projection step:

$$\tilde{x}_{t-1} \sim p(x_{t-1} \mid x_t), \quad x_{t-1} = \Pi_{\mathcal{C}}(\tilde{x}_{t-1}), \qquad (1)$$

where the projection operator $\Pi_{\mathcal{C}}(\tilde{x}) = \arg\min_{x \in \mathcal{C}} \|x - \tilde{x}\|^2$ maps any sample to its nearest feasible point.

This projection can be efficiently computed through differentiable optimization techniques (e.g., Lagrangian relaxation), achieving high constraint satisfaction in continuous diffusion models while maintaining sample quality close to the original diffusion output.

### 2.4. Motivation of NSPSG

Extending this projection-based method to discrete graph generation faces a fundamental challenge: the absence of gradients. Unlike continuous spaces where projection can be computed through gradient-based optimization, discrete graph states $G_t$ lack the differentiability required for such computation (Khalafi et al., 2024). This makes the projection method directly inapplicable to discrete domains.

We address this by reformulating the projection step as a discrete constraint satisfaction problem, where graph constraints are encoded symbolically and solved via automated reasoning. To further mitigate computational cost, we propose a neural projector that learns to approximate this reasoning process, enabling efficient batch projection.

## 3. Method

We propose NSPSG, a plug-and-play sampling approach that strictly enforces hard constraints within discrete diffusion models. Instead of relying on the diffusion model to learn constraints implicitly, we explicitly project invalid samples onto the feasible set during generation. As illustrated in Figure 1, we integrate a discrete projection operator into the denoising steps to achieve constrained graph generation (Section 3.1). We realize this projection operator by encoding constraints as logical formulas and solving them by automated reasoning (Section 3.2). To further accelerate sampling, a neural projector is trained to approximate this symbolic reasoning process (Section 3.3).

### 3.1. Discrete-Projected Diffusion Framework

Our framework combines diffusion sampling with discrete-projected constraint enforcement. Specifically, at denoising timestep $t$, we sample an unconstrained graph $\tilde{G}_{t-1}$ and then project it onto the feasible set $\mathcal{C}$ with a projection operator $\Pi_{\mathcal{C}}$:

$$\tilde{G}_{t-1} \sim q(G_{t-1} \mid G_t, \hat{G}_0), \quad G_{t-1} = \Pi_{\mathcal{C}}(\tilde{G}_{t-1}), \quad (2)$$

where $\Pi_{\mathcal{C}}(\tilde{G}) = \arg\min_{G \in \mathcal{C}} D(\tilde{G}, G)$ and $D$ is the graph distance metric detailed below.

An important consideration in practice is when to apply the projection operator $\Pi_{\mathcal{C}}$ during denoising. In continuous domains, methods (Christopher et al., 2024; Liang et al., 2025; Khalafi et al., 2024) demonstrate that projection throughout all denoising steps outperforms projecting only at the last step, as it prevents large distributional shifts by ensuring the distance to the feasible space decreases monotonically. However, we observe that this strategy may not always extend to discrete graph generation. Through empirical analysis on a representative scenario (Appendix A), we find that full projection can accumulate substantial distributional drift and may not always be the optimal choice. Our experiments in Experiment 4.2 further corroborate our analysis.

### 3.2. Exact Projection via Symbolic Reasoning

We achieve exact projection by formulating it as a symbolic reasoning task. Given a sampled graph $\tilde{G}$, the projection operator $\Pi_{\mathcal{C}}$ finds a feasible graph that satisfies all constraints in $\mathcal{C}$ (hard constraints) while minimizing modifications to $\tilde{G}$ (soft constraints). We employ SMT solvers to solve this task. Hard constraints are categorized into two types based on their computational properties: Arithmetic constraints(e.g., number of edges, degree bounds) that admit efficient SMT encodings (Sharma et al., 2024; Li et al., 2024) and structural constraints(e.g., connectivity, planarity) that require counterexample-guided refinement (Henzinger et al., 2003) as direct SMT encoding is computationally intractable.

**Solving Arithmetic Constraints.** Arithmetic constraints on graphs can be uniformly expressed as relations $\mathcal{R}(g_i(X, E))$, where $g_i$ is a real-valued function, and $\mathcal{R}$ represents relational operators such as equality, inequality, or Boolean combinations. These constraints can be systematically translated into SMT formulas for efficient solving.

To enable uniform constraint solving, we declare graph components as discrete decision variables (Reynolds et al., 2016; Balunovic et al., 2018), i.e., $\bar{X} = (\bar{x}_i)_{1 \leq i \leq n}$ and $\bar{E} = (\bar{e}_{ij})_{1 \leq i,j \leq n}$, where each variable is assigned to an atom in the SMT solver with domains defined by categorical spaces. Then, for any finitely definable constraint (Bansal et al., 2018), we introduce auxiliary variables aux to reformulate constraint $g(X, E)$ into an equivalent SMT-

compatible formula $g'(\bar{X}, \bar{E}, \text{aux})$ with the same relational operator.

We then define the complete constraint formula as $\Phi(\bar{X}, \bar{E}, \text{aux}) := \varphi_x \wedge \varphi_e \wedge \varphi_{aux} \wedge \varphi_g$, where $\varphi_x$ and $\varphi_e$ encode domain constraints, $\varphi_{aux}$ defines auxiliary variables, and $\varphi_g = \bigwedge_k \mathcal{R}_k(g'_k(\bar{X}, \bar{E}, \text{aux}))$ encodes all graph constraints. This formulation enables SMT solvers to handle a broad class of graph constraints(e.g., non-differentiable or non-linear) (Sharma et al., 2024; Li et al., 2024). Complete details and examples are provided in Appendix B.

**Solving Structural Constraints.** Structural constraints govern the global topological properties of graphs (Madeira et al., 2024; Kong et al., 2023) . Directly encoding these properties into SMT formulas is challenging because they involve complex disjunctive conditions over the graph structure. For example, ensuring acyclicity requires that for every potential cycle, at least one edge must be absent. Such encodings introduce a large number of complex clauses and auxiliary variables, resulting in formulas that are prohibitively expensive for SMT solvers to process.

We address this challenge by exploiting a key insight. While solving graphs that satisfy structural constraints is hard, verifying whether a given graph satisfies these constraints and identifying violations can often be done efficiently. Many structural properties admit polynomial-time verification procedures (Hopcroft & Tarjan, 1974; Basu & Pramanik, 2023) that not only check satisfaction but also return concrete counterexamples when violations occur.

We then adopt a CEGAR strategy (Counter Example-Guided Abstraction Refinement) (König et al., 2025; Kreft et al., 2023). Specifically, we iteratively solve the current constraint set with an SMT solver to obtain a candidate graph and then verify the candidate using a dedicated checker. If violations exist, we extract the counterexample and translate it into blocking constraints, and add these constraints to prevent the same violation in future iterations. This process continues until no counterexamples are found. By incrementally adding only necessary blocking constraints rather than encoding all structural requirements upfront, we achieve a practical and scalable solution. Detailed algorithm and analysis are provided in Appendix C.

**Encoding Soft Constraints.** Soft constraints ensure that projected graphs remain close to the original sampled graphs $\tilde{G}$, thereby preserving fidelity to the learned distribution. We achieve this by encoding a graph distance (Gao et al., 2010) as the optimization objective in MaxSMT, defined as:

$$D(G, \tilde{G}) = \frac{1}{n} \sum_{i=1}^{n} [x_i \neq \tilde{x}_i] + \frac{2}{n(n-1)} \sum_{1 \leq i < j \leq n} |e_{ij} - \tilde{e}_{ij}|,$$
$$(3)$$

where $[\cdot]$ denotes the indicator function.

---

**Algorithm 1** Sampling Algorithm

---

1: **Input:** Constraints $C$ (split into arithmetic $C_{arith}$ and structural $C_{struct}$), Diffusion Model $\mathcal{M}$
2: **Output:** Generated Graph $G_0$
3: Sample $G_T \sim q_X(n) \times q_E(n)$ {Random graph}
4: Declare decision variables $\bar{x}_i :$ Int and $\bar{e}_{ij} :$ Bool
5: Use Section 3.2 to construct $\Phi_{\text{arith}}$ with $\bar{X}, \bar{E}$
6: **for** $t = T$ **to** 1 **do**
7:    $\hat{G}_0 = \mathcal{M}(G_0 \mid G_t)$ {Get the clean data}
8:    $p(G_{t-1} \mid G_t) \approx q(G_{t-1} \mid G_t, \hat{G}_0)$ {Posterior}
9:    $\tilde{G}_{t-1} \sim p(G_{t-1} \mid G_t)$ {Categorical Distribution}
10:    **if** $t \leq T_0$ {Project, e.g., $T_0 = 0.05T$} **then**
11:      Construct Soft Constraint $\Phi_{\text{soft}}$
12:      **if** $C_{struct} \neq \emptyset$ **then**
13:        $(\text{status, model}) \leftarrow \text{Alg2}(C_{\text{struct}}, \Phi_{\text{arith}}, \Phi_{\text{soft}})$
14:      **else**
15:        $(\text{status, model}) \leftarrow \text{MaxSMT}(\Phi_{\text{arith}}, \Phi_{\text{soft}})$
16:      **end if**
17:      **if** status $=$ SAT **then**
18:        $G_{t-1} \leftarrow \text{Extract\_Graph(model)}$
19:      **else**
20:        $G_{t-1} \leftarrow \tilde{G}_{t-1}$
21:      **end if**
22:    **else**
23:      $G_{t-1} \leftarrow \tilde{G}_{t-1}$
24:    **end if**
25: **end for**
26: **return** $G_0$

---

With this, the discrete projector $\Pi_C$ returns a hard-feasible graph that minimizes the distance to the diffusion sample:

$$\Pi_{\mathcal{C}}(\tilde{G}) = \arg\min_G D(G, \tilde{G}) \quad \text{s.t.} \quad \Phi_{\text{hard}}(G). \quad (4)$$

SMT encoding can conveniently accommodate soft constraints, as it is also an arithmetic expression, so that we can obtain the corresponding constraint representation $\Phi_{\text{soft}}(\bar{X}, \bar{E}, \text{aux})$, add it as soft constraints to the solver, and try to minimize it. This MaxSMT formulation yields graphs that strictly satisfy all hard constraints while remaining close to the learned distribution, thus preserving the quality of generated samples. The whole sampling process is shown in Algorithm 1, where Alg2 is detailed in Algorithm 2.

### 3.3. Efficient Projection via Neural Approximation

To achieve efficient projection, we introduce an auto-regressive neural projector to approximate the behavior of symbolic solvers (Feng et al., 2024; Asthana et al., 2024). Although symbolic solvers guarantees exactness, it is computationally expensive and inherently sequential, making it time-consuming to generate large numbers of graphs. We address this by adopting a supervised learning approach, training the neural projector on invalid and valid graph pairs

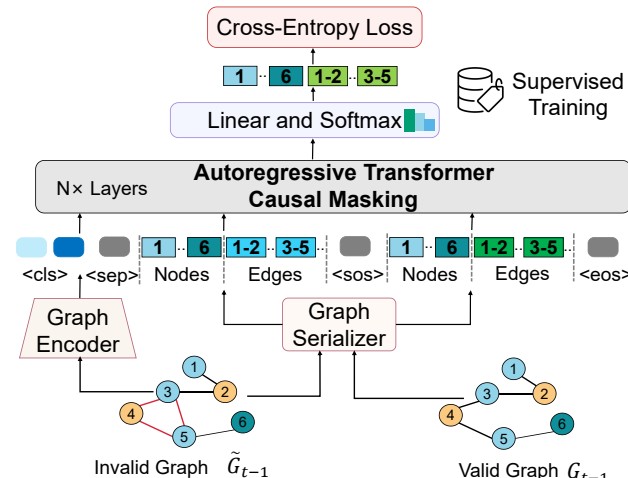

*Figure 2.* Training an Auto-regressive Neural Projector. The model uses a transformer decoder to generate the valid graph $G_{t-1}$ based on the invalid graph $\tilde{G}_{t-1}$. The input sequence consists of global context and the serialized invalid graph, from which the model learns to predict the valid graph autoregressively.

generated by the SMT solver (illustrated in Figure 2).

**Training Data Generation.** We first perform offline correction of diffusion samples $\{\tilde{G}_i\}_{i=1}^M$ using the SMT solver to obtain training pairs $\mathcal{D} = \{(\tilde{G}_i, \hat{G}_i)\}_{i=1}^M$, where $\hat{G}_i = P_{\text{SMT}}(\tilde{G}_i)$ denotes the SMT projection.

**Auto-Regressive Architecture Design.** We formulate the projection task as a conditional generation problem and employ an auto-regressive Transformer to model the mapping from invalid to valid graphs (Kong et al., 2023; Chen et al., 2025b). The architecture consists of two key components. First, a graph encoder embeds the sampled graph $\tilde{G}_t$ into a global context representation that captures topological features. Second, $\tilde{G}_t$ is serialized into a linear token sequence containing node attributes and edge indicators, forming a source sequence that enables the model to perform direct corrections at the token level.

The model learns to generate the valid graph sequence conditioned on the invalid input. Let $s(\tilde{G}_t) = [s_1, \ldots, s_m]$ and $s(G_t) = [t_1, \ldots, t_n]$ denote the serialized sequences of the invalid and valid graphs. The Transformer takes as input both the graph encoding $\text{Enc}(\tilde{G}_t)$ and the source sequence $s(\tilde{G}_t)$, and is trained to predict the target sequence $s(G_t)$:

$$\mathcal{L}(\theta) = -\frac{1}{M} \sum_{i=1}^M \sum_{l=1}^{n_i} \log p_\theta(t_l^i \mid \text{Enc}(\tilde{G}_t^i), s(\tilde{G}_t^i), t_{<l}^i),$$
$$(5)$$

where $M$ is the number of training samples, $n_i$ is the sequence length for sample $i$, and $t_{<l}^i$ denotes all previously generated tokens up to position $l$.

At inference time, given an invalid graph $\tilde{G}_t$, we feed its

graph embedding and serialized structure into the Transformer. The model then autoregressively generates the corrected which is deserialized into the valid graph $G_t$, enabling efficient projection. More implementation details are provided in Appendix D.

# 4. Experiments

We conducted extensive experiments to validate the capability of NSPSG to generate constraint-satisfying graphs across diverse scenarios. **(1) Basic Constraints.** We evaluate NSPSG on fundamental arithmetic and structural constraints separately (Experiments 4.1 and 4.2). **(2) Non-linear Arithmetic Constraints.** We test NSPSG on complex non-linear arithmetic constraints (Experiment 4.3 and Appendix E.4). **(3) Combined Constraints.** We examine scenarios where structural and arithmetic constraints must be satisfied simultaneously (Appendix E.5). **(4) Molecular Generation.** We apply NSPSG to molecule generation with chemical validity constraints (Appendix E.6).

## 4.1. Arithmetic Constraints On Real-World Graphs

**Setup.** We evaluate NSPSG on solving arithmetic constraints, following the experiment set of PRODIGY (Sharma et al., 2024). Datasets include three non-attributed real-world graphs: Community-small, Ego-small and EN-ZYMES (Jo et al., 2022). We use the same constraints of edge count, triangle count, and degree, which are generic arithmetic constraints on these non-attributed graphs.

*Table 1.* Average MMD scores for arithmetic constraints on real-world graphs, results for base diffusion models are also included.

| | | Community-small | | Ego-small | | Enzymes | |
|---|---|---|---|---|---|---|---|
| | | Avg.↓ | VAL$_c$ ↑ | Avg.↓ | VAL$_c$ ↑ | Avg.↓ | VAL$_c$ ↑ |
| Edge Count | EDP-GNN | 0.18 | 0.43 | 0.23 | 0.23 | 0.08 | 0.56 |
| | +PRODIGY | 0.26 | 0.52 | 0.02 | 0.64 | 0.08 | 0.95 |
| | +NSPSG (SMT) | **0.11** | **1.00** | **0.01** | **1.00** | **0.02** | **1.00** |
| | GDSS | 0.19 | 0.30 | 0.27 | 0.18 | 0.21 | 0.05 |
| | +PRODIGY | 0.21 | **1.00** | 0.05 | 0.62 | 0.54 | 0.82 |
| | +NSPSG (SMT) | **0.07** | **1.00** | **0.01** | **1.00** | **0.12** | **1.00** |
| | DruM | 0.42 | 0.25 | 0.40 | 0.10 | 0.16 | 0.29 |
| | +PRODIGY | 0.31 | 0.55 | 0.11 | 0.65 | 0.19 | 0.77 |
| | +NSPSG (SMT) | **0.23** | **1.00** | **0.09** | **1.00** | **0.11** | **1.00** |
| Triangle Count | EDP-GNN | 0.18 | 0.70 | 0.05 | 0.66 | 0.08 | 0.64 |
| | +PRODIGY | 0.20 | 0.83 | 0.03 | 0.98 | 0.39 | **1.00** |
| | +NSPSG (SMT) | **0.08** | **1.00** | **0.01** | **1.00** | **0.02** | **1.00** |
| | GDSS | 0.19 | 0.70 | 0.02 | 0.80 | 0.16 | 0.03 |
| | +PRODIGY | 0.14 | 0.90 | 0.06 | 0.82 | 0.13 | 0.96 |
| | +NSPSG (SMT) | **0.08** | **1.00** | **0.02** | **1.00** | **0.04** | **1.00** |
| | DruM | 0.42 | 0.30 | 0.12 | 0.48 | **0.02** | **1.00** |
| | +PRODIGY | 0.40 | 0.30 | 0.08 | 0.62 | 0.06 | **1.00** |
| | +NSPSG (SMT) | **0.21** | **1.00** | **0.03** | **1.00** | 0.03 | **1.00** |
| Degree | EDP-GNN | 0.18 | 0.55 | 0.12 | 0.36 | 0.08 | 0.52 |
| | +PRODIGY | 0.16 | 0.66 | **0.03** | 0.73 | 0.08 | **1.00** |
| | +NSPSG (SMT) | **0.07** | **1.00** | **0.03** | **1.00** | **0.06** | **1.00** |
| | GDSS | 0.19 | 0.60 | 0.13 | 0.32 | **0.14** | 0.40 |
| | +PRODIGY | 0.17 | **1.00** | 0.09 | 0.65 | 0.36 | **1.00** |
| | +NSPSG (SMT) | **0.09** | **1.00** | **0.02** | **1.00** | 0.18 | **1.00** |
| | DruM | 0.42 | 0.25 | 0.23 | 0.20 | 0.16 | 0.21 |
| | +PRODIGY | 0.40 | 0.25 | 0.08 | 0.55 | **0.07** | 0.80 |
| | +NSPSG (SMT) | **0.31** | **1.00** | **0.07** | **1.00** | 0.08 | **1.00** |

**Metrics.** We evaluate constraint satisfaction and distributional fidelity. For the satisfaction of constraints, we report the proportion of graphs generated that meet the specified constraint set $C$: $\mathrm{VAL}_C\big(\{G_i\}\big) := \frac{1}{N}\sum_{i=1}^{N}\mathbf{1}\big[G_i \in C\big]$, where $\{G_i\}_{i=1}^{N}$ denotes the set of graphs generated. To assess distributional fidelity, we measure how close the generated graphs are to the test dataset. Following PRODIGY, we quantify closeness via the Maximum Mean Discrepancy (MMD) (You et al., 2018) and report the average of three MMDs: degree, clustering coefficient, and orbit count, where lower values indicate better alignment.

**Results.** Table 1 summarizes the results. NSPSG outperforms PRODIGY and the unconditional diffusion models in these datasets, achieving 100% constraint satisfaction across all scenarios. NSPSG also achieves better MMDs than the original diffusion models and PRODIGY in most settings, demonstrating superior preservation of the learned distributional properties. Note that MMDs are computed against the test graphs that satisfy the constraints and all generated samples. Many unconditionally generated samples do not meet the constraints and deviate from the target distribution. In contrast, the SMT-based projection enforces feasibility while minimally altering the samples, yielding projected graphs that remain close to the learned distribution. More details and running time are reported in Appendix E.2.

## 4.2. Structural Constraints On Synthetic Graphs

**Setup.** We evaluate NSPSG on structural constraints, following Construct (Madeira et al., 2024). We consider three synthetic, unattributed datasets with distinct structural properties: Planar (Martinkus et al., 2022), Tree (Bergmeister et al., 2023) and Lobster (Liao et al., 2019).

**Metrics.** Following (Bergmeister et al., 2023), we evaluate both distributional fidelity and structural validity. For distributional similarity, we compute MMD over five graph descriptors (degree, clustering coefficient, orbit count, spectral, and wavelet) and report their average Ratio, where smaller values indicate better performance. For structural validity, we report three set-level measures: Unique, Novel, and Valid. We summarize these via V.U.N. We also report Property scores for specific constraints.

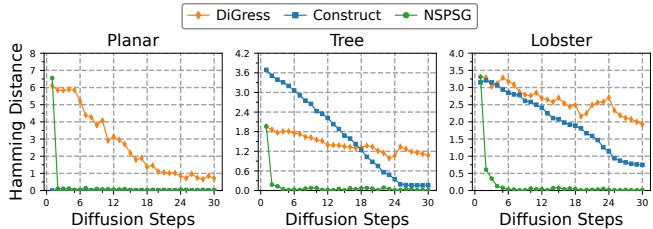

*Figure 3.* Hamming Distance for these datasets. The horizontal axis represents the starting timestep of the final 30 steps in the reverse diffusion sampling process.

*Table 2.* Graph generation performance on synthetic graphs with structural constraints. More results can be found in (Madeira et al., 2024).

| Model | Deg. ↓ | Clus. ↓ | Orbit ↓ | Spec. ↓ | Wavelet ↓ | Valid ↑ | Unique ↑ | Novel ↑ | V.U.N. ↑ | Property ↑ |
|---|---|---|---|---|---|---|---|---|---|---|
| **Planar Dataset** | | | | | | | | | | |
| Train set | 0.0002 | 0.0310 | 0.0005 | 0.0038 | 0.0012 | 100 | 100 | 0.0 | 0.0 | 100 |
| SPECTRE | 0.0005 | 0.0785 | 0.0012 | 0.0112 | 0.0059 | 25.0 | 100 | 100 | 25.0 | — |
| DiGress | 0.0007 | 0.0780 | 0.0079 | 0.0098 | 0.0031 | 77.5 | 100 | 100 | 77.5 | — |
| HSpec (one-shot) | 0.0003 | **0.0245** | 0.0006 | 0.0104 | 0.0030 | 67.5 | 100 | 100 | 67.5 | — |
| GruM | 0.0004 | 0.0301 | **0.0002** | 0.0104 | 0.0020 | - | - | - | 90.0 | — |
| CatFlow | 0.0003 | 0.0403 | 0.0008 | - | - | - | - | - | 80.0 | — |
| G2PT$_{base}$ | 0.0018 | 0.0047 | 0.00 | 0.0081 | 0.0051 | — | — | — | **100** | — |
| DiGress+ | 0.0008 ± 0.0001 | 0.0410 ± 0.0033 | 0.0048 ± 0.0004 | 0.0056 ± 0.0004 | 0.0020 ± 0.0002 | 76.4 ± 1.3 | 100.0 ± 0.0 | 100.0 ± 0.0 | 76.4 ± 1.3 | 76.4 ± 1.3 |
| DisCo | **0.0002** ± 0.0001 | 0.0403 ± 0.0155 | 0.0009 ± 0.0004 | - | - | 83.6 ± 2.1 | 100.0 ± 0.0 | 100.0 ± 0.0 | 83.6 ± 2.1 | - |
| Cometh-PC | 0.0006 ± 0.0005 | 0.0434 ± 0.0093 | 0.0016 ± 0.0006 | **0.0049** ± 0.0008 | - | 99.5 ± 0.9 | 100.0 ± 0.0 | 100.0 ± 0.0 | 99.5 ± 0.9 | - |
| ConStruct | 0.0003 ± 0.0001 | 0.0403 ± 0.0047 | 0.0004 ± 0.0001 | 0.0053 ± 0.0004 | 0.0009 ± 0.0001 | **100.0** ± 0.0 | 100.0 ± 0.0 | 100.0 ± 0.0 | **100.0** ± 0.0 | 100.0 ± 0.0 |
| DeFoG | 0.0005 ± 0.0002 | 0.0501 ± 0.0149 | 0.0006 ± 0.0004 | 0.0072 ± 0.0011 | 0.0014 ± 0.0002 | 99.5 ± 1.0 | 100.0 ± 0.0 | 100.0 ± 0.0 | 99.5 ± 1.0 | - |
| NSPSG (SMT) | **0.0002** ± 0.0001 | 0.0374 ± 0.0001 | 0.0008 ± 0.0002 | **0.0049** ± 0.0002 | **0.0006** ± 0.0001 | **100.0** ± 0.0 | 100.0 ± 0.0 | 100.0 ± 0.0 | **100.0** ± 0.0 | 100.0 ± 0.0 |
| NSPSG (AR) | **0.0002** ± 0.0001 | 0.0387 ± 0.0003 | 0.0009 ± 0.0003 | 0.0060 ± 0.0006 | 0.0008 ± 0.0001 | **100.0** ± 0.0 | 100.0 ± 0.0 | 100.0 ± 0.0 | **100.0** ± 0.0 | 100.0 ± 0.0 |
| **Tree Dataset** | | | | | | | | | | |
| Train set | 0.0001 | 0.0000 | 0.0000 | 0.0075 | 0.0030 | 100 | 100 | 0.0 | 0.0 | 100 |
| DiGress | 0.0002 | 0.0000 | 0.0000 | 0.1113 | 0.0043 | 90.0 | 100 | 100 | 90.0 | — |
| BwR | 0.0016 | 0.1239 | 0.0003 | 0.0480 | 0.0388 | 0.0 | 100 | 100 | 0.0 | — |
| BigGG | 0.0014 | 0.0000 | 0.0000 | 0.0119 | 0.0058 | 100 | 87.5 | 50.0 | 75.0 | — |
| HSpec (one-shot) | 0.0004 | 0.0000 | 0.0000 | 0.0080 | 0.0055 | 82.5 | 100 | 100 | 82.5 | — |
| HSpec | **0.0001** | 0.0000 | 0.0000 | 0.0117 | 0.0047 | 100 | 100 | 100 | **100** | — |
| G2PT$_{base}$ | 0.0043 | 0.00 | 0.0001 | **0.0073** | 0.0057 | — | — | — | 99 | — |
| DiGress+ | 0.0002 ± 0.0001 | 0.0000 ± 0.0000 | 0.0000 ± 0.0000 | 0.0092 ± 0.0005 | 0.0032 ± 0.0001 | 91.6 ± 0.7 | 100.0 ± 0.0 | 100.0 ± 0.0 | 91.6 ± 0.7 | 97.0 ± 0.8 |
| ConStruct | 0.0003 ± 0.0001 | 0.0000 ± 0.0000 | 0.0000 ± 0.0000 | **0.0073** ± 0.0008 | 0.0034 ± 0.0002 | 83.0 ± 1.8 | 100.0 ± 0.0 | 100.0 ± 0.0 | 83.0 ± 1.8 | 100.0 ± 0.0 |
| DeFoG | 0.0002 ± 0.0001 | 0.0000 ± 0.0000 | 0.0000 ± 0.0000 | 0.0108 ± 0.0028 | 0.0046 ± 0.0004 | 96.5 ± 2.6 | 100.0 ± 0.0 | 100.0 ± 0.0 | 96.5 ± 2.6 | - |
| NSPSG (SMT) | 0.0002 ± 0.0001 | 0.0000 ± 0.0000 | 0.0000 ± 0.0000 | 0.0081 ± 0.0001 | **0.0031** ± 0.0001 | **100.0** ± 0.0 | 100.0 ± 0.0 | 100.0 ± 0.0 | **100.0** ± 0.0 | 100.0 ± 0.0 |
| NSPSG (AR) | 0.0002 ± 0.0001 | 0.0000 ± 0.0000 | 0.0000 ± 0.0000 | 0.0074 ± 0.0002 | 0.0035 ± 0.0001 | **100.0** ± 0.0 | 100.0 ± 0.0 | 100.0 ± 0.0 | **100.0** ± 0.0 | 100.0 ± 0.0 |
| **Lobster Dataset** | | | | | | | | | | |
| Train set | 0.0002 | 0.0000 | 0.0000 | 0.0070 | 0.0070 | 100 | 100 | 0.0 | 0.0 | 100 |
| GRAN | 0.038 | 0.000 | 0.001 | 0.027 | — | — | — | 88.0 | — | — |
| GraphGen | 0.548 | 0.040 | 0.247 | — | — | — | — | — | — | — |
| BiGG | 0.000 | 0.000 | 0.000 | 0.009 | — | — | 100 | — | — | — |
| GDSS | 0.117 | 0.002 | 0.149 | — | — | 18.2 | 100 | 100 | 18.2 | — |
| BwR | 0.316 | 0.000 | 0.247 | — | — | 100 | 63.6 | 100 | 63.6 | — |
| GEEL | 0.002 | 0.000 | 0.001 | — | — | 72.7 | 100 | 72.7 | ≤ 72.7 | — |
| HGGT | 0.003 | 0.000 | 0.015 | — | — | — | — | — | — | — |
| DiGress | 0.021 | 0.000 | 0.004 | — | — | 54.5 | 100 | 100 | 54.5 | — |
| G2PT$_{base}$ | 0.001 | 0.00 | 0.00 | **0.004** | 0.01 | — | — | — | **100** | — |
| DiGress+ | 0.0005 ± 0.0001 | 0.0000 ± 0.0000 | 0.0000 ± 0.0000 | 0.0114 ± 0.0006 | 0.0093 ± 0.0005 | 79.0 ± 1.1 | 98.0 ± 0.7 | 96.6 ± 0.6 | 69.4 ± 1.2 | 76.8 ± 1.7 |
| ConStruct | **0.0003** ± 0.0001 | 0.0000 ± 0.0000 | 0.0000 ± 0.0000 | 0.0092 ± 0.0009 | **0.0074** ± 0.0009 | 86.8 ± 2.4 | 98.8 ± 0.6 | 97.0 ± 0.9 | 83.2 ± 2.3 | 100.0 ± 0.0 |
| NSPSG (SMT) | 0.0008 ± 0.0001 | 0.0000 ± 0.0000 | 0.0000 ± 0.0000 | 0.0080 ± 0.0001 | 0.0078 ± 0.0001 | **100.0** ± 0.0 | 99.9 ± 0.0 | 98.3 ± 0.2 | 96.9 ± 0.0 | 100.0 ± 0.0 |
| NSPSG (AR) | 0.0008 ± 0.0001 | 0.0000 ± 0.0000 | 0.0000 ± 0.0000 | 0.0089 ± 0.0003 | 0.0081 ± 0.0003 | **100.0** ± 0.0 | 100.0 ± 0.0 | 100.0 ± 0.0 | **100.0** ± 0.0 | 100.0 ± 0.0 |

**Results.** Table 2 reports the performance of these datasets. NSPSG surpasses ConStruct and other baselines in the three datasets, achieving 100% Valid and 100% Property (both SMT projector and auto-regressive neural projector). Furthermore, compared to ConStruct, NSPSG can accommodate a broader range of structural constraints, such as connectedness, which does not align with the constraint property defined in ConStruct. By employing a verifier that returns explicit disconnected components and constructs counterexamples, the SMT solver is enabled to find a globally consistent correction that satisfies all these constraints.

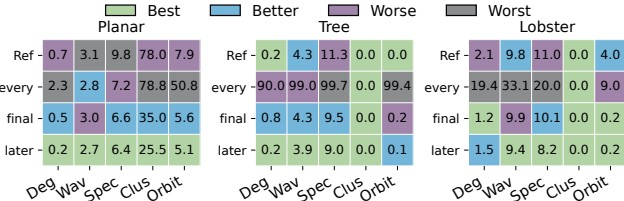

*Figure 4.* Data Distribution for these datasets. All values are scaled by $10^3$. Ref refers to the DiGress model without projection, later refers to DiGress with projection during the late 30 steps.

Figure 3 and Figure 4 present the results of different projection strategies. For unconditional sampling, the average feasibility distance $d_t$ remains moderate but does not converge to a small magnitude, indicating that relying solely on the diffusion model makes it challenging to generate data that strictly satisfy the constraints. Meanwhile, ConStruct's method also struggles to ensure the convergence of $d_t$, except for planarity since no more edges are added.

By contrast, applying projection in the late sampling steps via an SMT solver approximately guarantees the convergence of $d_t$ to near zero, inducing smaller distribution shifts in these steps and also achieve better results than DiGress without projection. This performance outperforms the projection performed solely in the final step, an observation consistent with previous studies (Liang et al., 2025; Sharma et al., 2024; Christopher et al., 2024). Notably, projection exclusively in the late steps incurs only a modest increase in sampling time, while projection at each sampling step not only induces larger shifts in the data distribution but also incurs higher computational cost. Additional experimental details, comprehensive runtime analysis, and scalability results for larger graphs are provided in Appendix E.3.

*Table 3.* Evaluation results on LP dataset. AR refers to the auto-regressive neural projector.

| Model | $\rho = 0$ | | | $\rho = 0.5$ | | | $\rho = 1$ | | |
| | Validity ↑ | $W_1$/MMD ↓ | | Validity ↑ | $W_1$/MMD ↓ | | Validity ↑ | $W_1$/MMD ↓ | |
| | | $L(\times 10^{-1})$ | $|\mathcal{V}^{(l)}|(\times 10^{-1})$ | | $L$ | $|\mathcal{V}^{(l)}|$ | | $L$ | $|\mathcal{V}^{(l)}|$ |
|---|---|---|---|---|---|---|---|---|---|
| D-VAE | $0.27 \pm 0.03$ | $8.7 \pm 1.0$ | $1.9 \pm 0.3$ | $0.37 \pm 0.04$ | $9.8 \pm 1.6$ | $1.9 \pm 0.6$ | $0.89 \pm 0.01$ | $8.8 \pm 0.9$ | $1.9 \pm 0.5$ |
| GraphRNN | $0.25 \pm 0.02$ | $9.8 \pm 0.2$ | $1.2 \pm 0.2$ | $0.34 \pm 0.07$ | $12.0 \pm 0.1$ | $1.8 \pm 0.2$ | $0.59 \pm 0.02$ | $14.0 \pm 1.0$ | $2.1 \pm 0.1$ |
| GraphPNAS | $0.23 \pm 0.04$ | $17.0 \pm 4.0$ | $2.2 \pm 0.7$ | $0.24 \pm 0.03$ | $20.0 \pm 3.0$ | $3.2 \pm 1.3$ | $0.67 \pm 0.04$ | $10.0 \pm 3.0$ | $0.8 \pm 0.6$ |
| OneShotDAG | $0.37 \pm 0.02$ | $6.4 \pm 0.9$ | $1.5 \pm 0.1$ | $0.31 \pm 0.07$ | $3.9 \pm 0.7$ | $1.3 \pm 0.0$ | $0.50 \pm 0.08$ | $4.1 \pm 2.4$ | $1.1 \pm 0.4$ |
| LayerDAG ($T = 1$) | $0.26 \pm 0.06$ | $1.6 \pm 0.8$ | $0.14 \pm 0.0$ | $0.36 \pm 0.02$ | $1.3 \pm 0.3$ | $0.12 \pm 0.1$ | $0.95 \pm 0.01$ | $2.0 \pm 0.1$ | $\mathbf{0.08 \pm 0.0}$ |
| LayerDAG | $0.56 \pm 0.02$ | $1.6 \pm 1.0$ | $\mathbf{0.10 \pm 0.0}$ | $0.63 \pm 0.00$ | $1.8 \pm 1.1$ | $0.06 \pm 0.0$ | $0.96 \pm 0.02$ | $1.9 \pm 0.6$ | $0.10 \pm 0.3$ |
| NSPSG (SMT) | $\mathbf{0.99 \pm 0.00}$ | $1.4 \pm 0.9$ | $\mathbf{0.10 \pm 0.0}$ | $\mathbf{0.99 \pm 0.00}$ | $1.2 \pm 0.3$ | $\mathbf{0.05 \pm 0.0}$ | $\mathbf{1.00 \pm 0.00}$ | $1.4 \pm 0.1$ | $\mathbf{0.08 \pm 0.0}$ |
| NSPSG (GNN) | $0.76 \pm 0.01$ | $2.3 \pm 1.2$ | $0.14 \pm 0.1$ | $0.79 \pm 0.03$ | $2.7 \pm 0.2$ | $0.10 \pm 0.4$ | $0.96 \pm 0.00$ | $1.8 \pm 0.7$ | $0.13 \pm 0.1$ |
| NSPSG (AR) | $0.97 \pm 0.01$ | $\mathbf{1.4 \pm 1.1}$ | $\mathbf{0.10 \pm 0.0}$ | $0.98 \pm 0.00$ | $1.3 \pm 1.0$ | $0.08 \pm 0.0$ | $0.99 \pm 0.00$ | $1.7 \pm 0.3$ | $0.11 \pm 0.0$ |

In summary, NSPSG delivers strict structural feasibility with minimal modification, maintains or improves distributional fidelity, outperforming the current SOTA model for constrained generation under structural properties.

### 4.3. Non-Linear Constraints On Attributed Graphs

**Setup.** To demonstrate the capability of NSPSG on more complex domains, we evaluate it on directed acyclic graphs (DAGs) with logical rules, following LayerDAG (Li et al., 2024). We use the synthetic LP benchmark as hard constraint which requires each node's predecessors to have approximately equal numbers of attribute 0 and attribute 1, with strictness controlled by parameter $\rho \in \{0, 0.5, 1\}$.

**Metrics.** We evaluate the hard logical constraint by reporting the validity rate, namely the proportion of DAGs generated that satisfy the hard rule. To assess distributional alignment, we compare statistics of the generated DAGs with the test dataset. Specifically, we compute the 1-Wasserstein distance ($W_1$) between the distributions of the number of layers $L$ and the MMD between the distributions of layer sizes $|\mathcal{V}^{(l)}|$ (You et al., 2018).

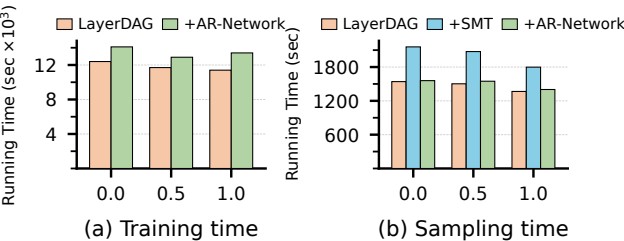

*Figure 5.* Training and sampling time for LP dataset. The training time includes the time for SMT-based training data generation.

**Results.** Table 3 summarizes the evaluation on LP dataset and Figure 5 reports the running time. NSPSG markedly outperforms LayerDAG and other baselines in satisfying the logical balance constraint, with better improvement under stricter rules (smaller $\rho$). Using SMT as projector, NSPSG achieves 99–100% satisfaction in all settings. Simulating

the projector via an auto-regressive neural projector achieves nearly the same validity as the SMT solver while substantially reducing generation time, approximately close to LayerDAG. More generally, LP constraints are non-linear, making unsupervised penalty-based training ineffective (Sharma et al., 2024). Unlike prior works, NSPSG can address this issue and significantly enhance accuracy with supervised training. Further studies indicate the effectiveness of auto-regressive neural projector. The replacement with a GNN-based backbone model struggles to capture long-range dependencies among layers and attribute balances, yielding accuracy clearly below both SMT and the auto-regressive neural projector. Additional constraint descriptions, training details, more challenging experiments and analysis on neural projector are provided in Appendix E.4.

For distribution metrics, NSPSG matches or surpasses LayerDAG and other baselines on $W_1$ and MMD in most cases, demonstrating that our method can both strictly enforce logical rules and maintain high fidelity to the target distribution.

## 5. Related Work

Diffusion models based on iterative noising and denoising have achieved impressive results in multiple domains, such as images (Moser et al., 2024; Corneanu et al., 2024; Huang et al., 2025) and videos (Croitoru et al., 2023; Xing et al., 2024). Recently, this paradigm has been introduced to graph generation (Niu et al., 2020; Huang et al., 2024; Chen et al., 2023; Asthana et al., 2024), often in a one-shot setting that samples an entire graph structure at once, delivering strong performance in modeling graph distributions and producing high-quality samples. To better accommodate graph-specific characteristics, researchers have designed graph-specific diffusion models, such as score-based approaches (Niu et al., 2020; Jo et al., 2022; Yan et al., 2023) and discrete diffusion process (Vignac et al., 2022; Huang et al., 2023; Xu et al., 2024; Tseng et al., 2023).

However, due to the inherent stochasticity of diffusion process, even when all training graphs are feasible, uncon-

ditional generation cannot guarantee that sampled graphs satisfy user-specified constraints (Liang et al., 2025; Christopher et al., 2024; Zampini et al., 2025; Cardei et al., 2025). This limitation has motivated constrained graph generation (Sharma et al., 2024; Madeira et al., 2024; Chen et al., 2023; You et al., 2018), which seeks to explicitly incorporate the prior knowledge about specific properties into generation process so that the samples can respect the required properties. Various diffusion-based approaches to this challenge can be broadly categorized by how they intervene.

**Conditional Generation.** These methods try to modify the model during training to bias samples toward feasibility, which generally adopt conditional diffusion via classifier guidance (Tenorio et al., 2025; Mercatali et al., 2024) or classifier-free guidance (Shenoy et al., 2024). ReClassifier guidance tries to train an auxiliary network to predict whether the current noisy sample $\tilde{G}_t$ satisfies the constraint and steers the reverse process using a gradient-based update, while classifier-free guidance instead learns a joint model that supports both conditional and unconditional sampling, as a result, each condition requires its own model and introducing a new control generally requires retraining. However, these methods provide only soft constraints and cannot guarantee satisfaction, while typically require large amounts of labeled data tailored to the target constraints, which is costly or impractical to obtain in structured domains.

**Designed Generation.** Beyond conditioning-based guidance, these works design bespoke diffusion models and posterior sampling so that the entire reverse process remains within the feasible set. For example, EDGE (Chen et al., 2023) enforces a hard constraint via degree guidance. CoCo-Graph (Ruiz-Botella et al., 2025) develops a molecular diffusion model tailored to chemical valence so that generated molecules are valid by construction. ConStruct (Madeira et al., 2024) targets edge-deletion–invariant properties with an edge-absorbing noise model and an edge-addition projector. While effective on their target properties, these designs are inherently specialized and difficult to extend. In practice, they do not readily support additional or heterogeneous constraints.

**Projected Generation.** This class of work, inspired by Mirrored Langevin Dynamics (Bubeck et al., 2018; Srinivasan et al., 2024), keeps the pretrained diffusion model fixed and projected the intermediate samples (Liang et al., 2025; Christopher et al., 2024). A representative instance is PRODIGY (Sharma et al., 2024), which relaxes adjacency matrices and categorical node features into continuous spaces and computes a Lagrangian-based projection toward the constraint set to guide sampling in each reverse step. Although efficient, the approach offers only approximate feasibility and cannot guarantee that the discretized outputs satisfy the constraints after rounding or threshold-

ing. Moreover, they are typically limited to relatively simple arithmetic constraints. Extending them to non-linear logical formulas or structural properties is challenging.

In contrast, NSPSG employs an SMT-based solver to project samples into the feasible set in the discrete domain, guaranteeing that the generated graphs strictly satisfy the specified constraints. With nodes, edges, attributes treated as integer or Boolean decision variables, and heterogeneous constraints encoded directly as logical and arithmetic predicates, NSPSG naturally handles diverse arithmetic expressions and structural predicates, which can even scale to combinations of constraints. In addition, SMT can efficiently generate labeled pairs, which we leverage to supervise a neural projector for batching generation at inference time. In general, NSPSG is a plug-and-play framework that combines with arbitrary unconditional diffusion backbones and does not require retraining diffusion model when constraints change.

# 6. Discussion and Conclusion

In this paper, we introduce NSPSG, a novel framework that explicitly injects domain constraints into graph diffusion models through discrete projection. Our approach addresses the fundamental challenge of constrained graph generation by exploiting the discrete structure of graphs, which allows hard constraints to be formulated as symbolic reasoning problems. Using an SMT-based projector and a neural projector, NSPSG ensures that outputs are realistic and strictly constraint-compliant. Experiments demonstrate that NSPSG achieves near optimal performance across various scenarios.

These contributions alleviate long-standing challenges in constrained graph generation, and open opportunities for further development in constraint-critical applications. Future work may extend our framework to broader domains such as network configuration (Schneider et al., 2021; El-Hassany et al., 2018), where both guaranteed constraint satisfaction and computational efficiency are essential.

## Acknowledgements

This work was supported in part by the National Natural Science Foundation of China under Grants (62471055, 62321001, U23B2001), the High-Quality Development Project of the MIIT(2440STCZB2584), the Ministry of Education and China Mobile Joint Fund (MCM20200202, MCM20180101), the Fundamental Research Funds for the Central Universities (2024PTB-004).

## Impact Statement

This paper presents NSPSG whose goal is to advance the field of machine learning, specifically in constrained graph generation. Our framework ensures that generated graphs

strictly satisfy specified constraints, which has important positive implications for safety-critical applications. By preventing generation of invalid or unsafe structures, our work makes generative models safer and more suitable for deployment in constraint-critical applications. We believe our contributions can enhance the reliability and trustworthiness of AI systems.

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

# A. Projection Strategy Analysis

We conduct an analysis of a representative and widely applicable scenario where the generation task involves only edge additions and deletions (e.g., arithmetic constraints like degree bounds or structural constraints like planarity) (Madeira et al., 2024; Chen et al., 2023). Through this analysis, we demonstrate that applying projection throughout the entire denoising process can be suboptimal. In contrast, concentrating the projection in late denoising steps reduces the expected feasibility distance while preserving distributional fidelity, thereby achieving superior performance.

For this analysis, we represent the edge configuration of a graph as a binary vector in the state space $\{0,1\}^m$, where $m$ denotes the number of edges. We use the Hamming distance $\mathrm{Ham}(x,y) = \sum_{i=1}^m \mathbf{1}\{x_i \neq y_i\}$ as the distance metric. The non-empty feasible set is denoted $\mathcal{C} \subseteq \{0,1\}^m$, and $c : \{0,1\}^m \to \mathcal{C}$ represents the deterministic projection that returns a nearest feasible configuration under Hamming distance with a fixed tie-breaking rule.

We consider a diffusion model $\mathcal{M}$ well-trained on constraint-satisfying data, which can generate samples that nearly satisfy the constraints and output the clean graph $X_0$ associated with a given noisy state $X_t$. The single-step posterior of $X_{t-1}$ can thus be written as $q_{t-1}(\cdot \mid X_t, X_0 = \mathcal{M}(X_t))$, where we omit explicit conditioning on $X_0$ for brevity.

Following the usual reverse-time indexing $t = T, T-1, \ldots, 1$, we denote the pre-projection distribution at step $t$ by $q_t$ and adopt a sample-then-project procedure:

$$X_t \sim q_t, \quad Z_t = c(X_t), \quad X_{t-1} \sim q_{t-1}(\cdot \mid Z_t), \tag{6}$$

where $Z_t$ is a deterministic function of $X_t$ and $X_{t-1}$ is sampled conditionally on $Z_t$.

## A.1. Ensuring Constraint Feasibility

Define the average feasibility distance as:

$$d_t := \mathbb{E}_{X_t \sim q_t}\big[\mathrm{Ham}\big(X_t, c(X_t)\big)\big]. \tag{7}$$

Fix an edge index $i \in \{1, \ldots, m\}$, and $z$ denote the projection value. We define the edge-wise inconsistency probability at step $t$ as:

$$\alpha_{t,i}(z) := \Pr(X_{t,i} \neq z_i \mid Z_t = z), \alpha_t = \mathbb{E}_{Z_t}\big[\mathbb{E}_i\big[\alpha_{t,i}(Z_t)\big]\big], \tag{8}$$

which measures how likely the bit $X_t$ disagrees with the feasible projection $Z_t$ at time $t$.

Since the Hamming distance is the sum of edge-wise mismatches and by linearity of expectation:

$$d_t = \mathbb{E}[\mathrm{Ham}(X_t, Z_t)] = \sum_{i=1}^m \mathbb{E}_{Z_t}\big[\Pr(X_{t,i} \neq z_i \mid Z_t = z)\big] = \sum_{i=1}^m \mathbb{E}_{Z_t}\big[\alpha_{t,i}(Z_t)\big] = m\,\mathbb{E}_{Z_t}\big[\mathbb{E}_i\big[\alpha_{t,i}(Z_t)\big]\big] = m\alpha_t, \tag{9}$$

so that controlling the edge-wise inconsistency probability $\alpha_t$ directly bounds the average feasibility distance $d_t$. To understand why applying projection during late denoising steps keeps $d_t$ small, we analyze the behavior of $\alpha_t$ in this regime. Our analysis reveals that two complementary factors jointly drive $\alpha_t$ to remain small in late steps, thereby ensuring that $d_t = m\alpha_t$ stays bounded.

**The samples remain stable in late diffusion steps.** As the reverse diffusion process progresses into late stages, the diffusion noise decreases substantially and approaches zero. Consequently, when $t$ is small, the forward diffusion adds minimal noise, making $X_t$ naturally close to $X_0$. For a well-trained diffusion model, the predictor's reconstruction $X_0 = \mathcal{M}(X_t)$ becomes highly accurate, yielding $X_0 \approx X_t$. This proximity implies that $X_t$ already closely resembles its clean target configuration and requires minimal further updates. For instance, in DiGress (Vignac et al., 2022), the one-step posterior for the discrete edge can be written in compact form as $q(X_{t-1} \mid X_t, X_0) \propto (X_t Q_t^\top) \odot (X_0 \bar{Q}_{t-1})$, where $Q_t$ is the forward transition matrix at step $t$, $\bar{Q}_{t-1} = Q_1 \cdots Q_{t-1}$, and $\odot$ denotes element-wise multiplication. In late stages, the forward step becomes nearly the identity ($Q_t \approx I$). Combined with $X_t \approx X_0$, this yields $X_{t-1} \approx X_t$, making edge flips rare and keeping $\alpha_t$ small.

**The projection guides sampling toward the feasible set.** The second factor is the guiding role of the projection operator. Training with data satisfying constraints biases the model towards feasible graphs (Mercatali et al., 2024), but this bias alone cannot ensure feasibility during generation. The key lies in our project-then-sample procedure. Rather than directly sampling

$X_{t-1}$ from $X_t$, we compute the feasible guide $Z_t = c(X_t)$ and condition the next step on $Z_t$, yielding $X_{t-1} \sim q_{t-1}(\cdot \mid Z_t)$. In late steps, $X_t$ is nearly clean and typically close to the feasible set, although randomness may introduce violations. By conditioning on the feasible $Z_t$, we exploit the model's preference for constraint-satisfying configurations. The conditional distribution $q_{t-1}(\cdot \mid Z_t)$ concentrates around $Z_t$, making $X_{t-1}$ likely to inherit feasibility while introducing only minor stochastic variations. This keeps $\alpha_t$ small and bounds $d_t$.

Together, these two factors ensure that the average feasibility distance $d_t$ decreases and converges to a small value in late denoising steps. Although it does not guarantee perfect constraint satisfaction at intermediate step, it explains why samples are effectively driven toward the constraint set in practice. When concentrated in the final steps, generated samples are more likely to satisfy the constraints, which is consistent with our experimental results in Experiment 4.2.

### A.2. Preserving Distributional Fidelity

While the projection operator $Z_t = c(X_t)$ guarantees constraint satisfaction by definition, a naive projection could trivially achieve this by mapping all inputs to a single valid configuration (i.e., mode collapse). Therefore, an important question is not merely whether validity improves, but whether the diversity and structural semantics of the learned distribution are preserved.

We quantify this by analyzing the Entropy Drop, $\Delta H_t = H(X_t) - H(Z_t)$, which measures the information loss induced by the projection. Since $Z_t$ is a deterministic function of $X_t$, the entropy drop is exactly equivalent to the conditional entropy:

$$H(X_t) - H(Z_t) = H(X_t \mid Z_t) \tag{10}$$

For this analysis, we use $h(u) = -u \log u - (1-u) \log(1-u)$ to denote the binary entropy function for $u \in (0, 1)$. We establish the following bound relating information loss to the projection magnitude: Let $d_t = \mathbb{E}[\mathrm{Ham}(X_t, Z_t)]$ be the expected Hamming distance between the pre-projection state $X_t$ and the post-projection state $Z_t$. The information loss is bounded by:

$$H(X_t \mid Z_t) \leq m \cdot h\left(\frac{d_t}{m}\right). \tag{11}$$

*Proof.* By definition, the conditional entropy is the expectation of the entropy of the conditional distribution: $H(X_t \mid Z_t) = \mathbb{E}_{z \sim q'_t}[H(X_t \mid Z_t = z)]$. Consider a fixed projection target $z \in \mathcal{C}$. For each coordinate $i \in \{1, \ldots, m\}$, let $p_{i,z} := \Pr(X_{t,i} \neq z_i \mid Z_t = z)$ be the error probability for bit $i$ given the target $z$. The entropy of the $i$-th bit conditioned on $z$ is exactly given by the binary entropy function $h(p_{i,z})$.

First, by the subadditivity of entropy (independence bound), the joint conditional entropy is upper-bounded by the sum of marginal entropies:

$$H(X_t \mid Z_t = z) \leq \sum_{i=1}^{m} H(X_{t,i} \mid Z_t = z) = \sum_{i=1}^{m} h(p_{i,z}). \tag{12}$$

Then, let $\delta_z := \sum_{i=1}^{m} p_{i,z} = \mathbb{E}[\mathrm{Ham}(X_t, z) \mid Z_t = z]$ denote the expected Hamming distance for the cluster mapping to $z$. Since $h(\cdot)$ is concave, we apply Jensen's inequality to the sum:

$$\sum_{i=1}^{m} h(p_{i,z}) = m \sum_{i=1}^{m} \frac{1}{m} h(p_{i,z}) \leq m \cdot h\left(\frac{1}{m} \sum_{i=1}^{m} p_{i,z}\right) = m \cdot h\left(\frac{\delta_z}{m}\right). \tag{13}$$

Finally, we substitute this back into the expectation over $z$ and apply Jensen's inequality a second time, leveraging the concavity of $h(\cdot)$ again:

$$H(X_t \mid Z_t) = \mathbb{E}_{z \sim q'_t}\left[H(X_t \mid Z_t = z)\right] \leq \mathbb{E}_{z \sim q'_t}\left[m \cdot h\left(\frac{\delta_z}{m}\right)\right] \leq m \cdot h\left(\mathbb{E}_{z \sim q'_t}\left[\frac{\delta_z}{m}\right]\right) = m \cdot h\left(\frac{d_t}{m}\right), \tag{14}$$

where the last equality follows from the law of total expectation: $\mathbb{E}_z[\delta_z] = d_t$. $\square$

This inequality provides an upper bound on the possible single-step increase in the Entropy Drop from the training data distribution. Based on the analysis in Section A.1 and the assumption that the training data satisfy the constraints, in the late

denoising steps, $d_t$ is not excessively large without projection and can be made particularly small with projection. This, coupled with the fact that the iterative sampling can mitigate distribution shifts to a certain extent (Cardei et al., 2025), leads to the conclusion that limiting projection to the late steps can not only ensure feasibility, but also yield a small bound on the total drift in KL divergence. In contrast, in the early steps of the process, the generated graphs contain substantial noise and exhibit a large deviation from the feasible region, so $d_t$ is large and trying to forcefully project these graphs into the feasible region would introduce a significant cumulative drift, which is difficult to rectify and also incurs substantial computational resource consumption. This analysis is also aligns with our experimental results in Experiment 4.2.

This analysis shows that for the aforementioned scenario, concentrating the projection in late denoising steps does not substantially increase the Entropy Drop while encouraging samples to satisfy the constraints under the stated assumptions. Importantly, this is a case-specific analysis rather than a universal theoretical result: it demonstrates that applying projection throughout the entire sampling process may not always be optimal, and more selective projection strategies can be beneficial.

## B. Arithmetic Constraints Solving Details

### B.1. Arithmetic Constraints Reformulation

In this work, we restrict the considered constraints to a finitely definable range (Bansal et al., 2018). The arithmetic constraint is any finite-length Boolean or arithmetic expression. Every primitive operator or function is natively supported by standard SMT theories or can be rewritten in finite steps to an equivalent expression over those atoms with the introduction of finite auxiliary variables (Reynolds et al., 2017; Babikian et al., 2020).

Under these assumptions, for any constraint $g$ that takes a graph as input, there exists a finite set of auxiliary variables aux and an equivalent formula $g'(\bar{X}, \bar{E}, \text{aux})$ composed solely of SMT-supported atoms, such that for any graph $G = (X, E)$ and relation symbol $\mathcal{R}$:

$$\mathcal{R}\big(g(X, E)\big) \implies \exists\, \text{aux} : \mathcal{R}\big(g'(\bar{X}, \bar{E}, \text{aux})\big). \tag{15}$$

Based on this, we define the constraint formula for SMT solving as $\Phi(\bar{X}, \bar{E}, \text{aux}) := \varphi_x \bigwedge \varphi_e \bigwedge \varphi_{aux} \bigwedge \varphi_g$, where

$$\begin{cases} \varphi_x = \bigwedge_i (\bar{x}_i = 1 \vee \bar{x}_i = 2... \vee \bar{x}_i = a) \\ \varphi_e = \bigwedge_{i,j} (\bar{e}_{i,j} = 0 \vee \bar{e}_{i,j} = 1) \\ \varphi_{aux}, \text{aux} = \text{Def}_{\text{aux}}(\bar{X}, \bar{E}) \\ \varphi_g = \bigwedge_k \mathcal{R}_k(g'_k(\bar{X}, \bar{E}, \text{aux})) \end{cases} \tag{16}$$

Then, we can unify a broad class of graph-based arithmetic hard constraints into $\Phi(\bar{X}, \bar{E}, \text{aux})$ and automatically solve them using the SMT solver. Leveraging background theories, SMT solvers can support constraints without requiring formulas to be differentiable, or linear(Sharma et al., 2024; Li et al., 2024; Madeira et al., 2024), thus expanding the range of constraints that can be solved.

### B.2. Arithmetic Constraints Solving Example

We then demonstrates our SMT encoding approach through a concrete example: constraining node degree to be at most $c$. We present a complete, self-contained formulation suitable for direct implementation in standard SMT solvers(e.g., Z3).

**Auxiliary Variable Introduction.** We begin by introducing auxiliary integer variables $\varphi_{aux} = \{\bar{d}_i\}_{i=1}^n$, where each $\bar{d}_i$ represents the degree of node $i$. These auxiliary variables bridge the gap between the Boolean edge representation and the arithmetic constraint on node degrees.

**Degree Computation.** For each node $i$, we define its degree $\bar{d}_i$ as the sum of the edge indicators incident. Let $\bar{e}_{ij}$ denote the integer interpretation of the Boolean edge indicator between nodes $i$ and $j$. We then assert the following equality constraint for every node $i$, as $\bar{d}_i = \sum_{j \neq i} \bar{e}_{ij}$, ensuring that $\bar{d}_i$ accurately counts the number of edges incident to node $i$.

**Degree Bound Enforcement.** Given a constant bound $c$, we impose the degree constraint on all nodes through the finite conjunction, as $\varphi_g = \bigwedge_{i=1}^n (\bar{d}_i \leq c)$, ensuring that no node in the generated graph exceeds the specified maximum degree.

**Implementation Using Z3.** To illustrate the practical implementation, we provide a minimal code snippet that instantiates these declarations and assertions in Z3, and the SMT solver can then directly solve these constraints to generate graphs satisfying the degree bounds:

```python
solver = z3.Solver()  # Declare SMT solver
# Declare Boolean edge variables for undirected graph (i < j)
e = {(i, j): Bool(f"e_{i}_{j}") for i in range(0, n) for j in range(i+1, n)}
# Declare integer auxiliary variables for degrees
d = {i: Int(f"d_{i}") for i in range(0, n)}
# For each node i, build incident-edge indicators, define degree, and constrain it
for i in range(0, n):
    degree_i = []
    for k in range(0, n):
        if k == i: continue
        a, b = (i, k) if i < k else (k, i)
        degree_i.append(If(e[(a, b)], 1, 0))
    # Define degree as sum of incident edges
    solver.add(d[i] == Sum(degree_i))
    # Enforce degree upper bound
    solver.add(d[i] <= c)
if solver.check() == z3.sat:
    model = solver.model()
    # Get the refined graph that satisfying constraints
```

## C. Structural Constraints Solving Details

### C.1. Incremental CEGAR

In practice, we perform structural projection using a CEGAR-style loop combined with a soft Hamming distance objective (detailed in Appendix A). A key property of this approach is monotonicity: the Hamming distance of candidates is non-decreasing across iterations. Specifically, if the MaxSMT solver returns an infeasible candidate at Hamming distance $d_j$, all subsequent candidates have Hamming distance $d_{j+1} \geq d_j$, since any solution with smaller Hamming distance would have been found earlier because the MaxSMT solver returns a candidate with minimal Hamming distance.

**Problem: Excessive iterations at the same Hamming distance.** Although Hamming distances increase monotonically across iterations, the solver may get stuck at a single distance level. At a fixed Hamming distance $d$, there can be many combinations of structural violations, particularly for dense constraints (e.g., planar graphs). The fundamental issue is that blocking clauses only explicitly forbid the specific counterexamples encountered, while other counterexamples may be satisfied implicitly in a given candidate without being enforced by constraints. When the solver searches for a new minimal Hamming distance candidate after adding a blocking clause, it may generate a solution that avoids the explicitly blocked violation but inadvertently reintroduces other structural violations that were previously satisfied. This occurs because the solver optimizes solely for minimal Hamming distance subject to the accumulated blocking clauses, without awareness of structural properties that were incidentally satisfied but not yet violated. Consequently, the solver may repeatedly return different infeasible candidates at the same Hamming distance, leading to excessive iterations without advancing to higher Hamming distance.

**Solution: Bounded attempts per Hamming distance level.** To address this problem, we limit iterations at each Hamming distance level to at most $k$ attempts. If no valid graph is found within $k$ iterations at distance $d$, we explicitly force the solver to move to distance $d + 1$ by adding the cardinality constraint $\sum_e \text{diff}_e \geq d + 1$, where $\text{diff}_e$ are edge-flip indicators relative to $X_t$. This constraint, together with accumulated blocking clauses, restricts the search space to higher Hamming distances. The parameter $k$ controls the trade-off: larger $k$ allows more thorough search at each level (better optimality), while smaller $k$ guarantees faster progression (better efficiency). The complete algorithm is detailed in Algorithm 2.

### C.2. Solving Efficiency Analysis

Following the analysis in Section 3.1 and Appendix A, we perform projection only in the late diffusion steps where samples are close to satisfying the constraints. We now discuss several key factors that contribute to the efficiency of our projection framework and demonstrate how these design choices enable practical deployment.

**Late projection with small overhead.** An important design is to apply projection only in the late stages of the diffusion process (e.g., projection on the late 5% diffusion steps). This strategy differs from approaches that require projection throughout the entire generation process, helping to reduce computational overhead while preserving generation quality.

---

**Algorithm 2** SMT Projection For Structural Constraints (we omit nodes types for simplicity)

---

1: **Input:** Graph $\tilde{G} = (\tilde{E})$, Structural Constraints $C$, Arithmetic Constraints $\Phi_{\text{arith}}$ if exist, Soft Constraints $\Phi_{\text{soft}}$ detailed in Algorithm 1 , Structural Verifier $\mathcal{V}$, Max times for Incremental CEGAR $K$
2: **Output:** A graph $\hat{G} = (\hat{E})$ satisfying all structural constraints $C$, or UNSAT
3: Initialize constraints $\Phi_0 \leftarrow \emptyset$
4: Declare auxiliary variables $\text{diff}_{ij} :$ Bool, DiffEdges : Int, use decision variables $\bar{e}_{ij} :$ Bool
5: Declare Hamming Distance $\text{diff}_{ij} := \text{If}(\bar{e}_{ij} \neq \tilde{e}_{ij}, 1, 0)$ and $\text{DiffEdges} := \sum_{(i,j)} \text{diff}_{ij}$
6: **for** $u \leftarrow 0, 1, 2, \ldots \max\_\text{iteration}$ **do**
7:     $(\text{status, model}) \leftarrow \text{MaxSMT}(\Phi_u, \Phi_{\text{arith}}, \Phi_{\text{soft}})$
8:     **if** $\text{status} = \text{UNSAT}$ **then**
9:         **return** UNSAT
10:     **end if**
11:     $(\hat{E}) \leftarrow \text{Extract\_Graph(model)}$ {Get the candidate graph.}
12:     $(\text{valid}, (\hat{S}_{\text{conflict}})) \leftarrow \mathcal{V}(\hat{E})$ {Check constraints. If invalid, return minimal conflict set $\hat{S}_{\text{conflict}}$.}
13:     **if** valid **then**
14:         **return** $(\text{status, model})$
15:     **else**
16:         $\psi_{\text{block}} \leftarrow \neg \left( \bigwedge_{(i,j) \in \hat{S}_{\text{conflict}}} (\bar{e}_{ij} = \hat{e}_{ij}) \right)$ {Blocking Clause: Prevent this specific violation pattern}
17:         $\Phi_{u+1} \leftarrow \Phi_u \wedge \psi_{\text{block}}$ {Add as a Hard Constraint}
18:         $d_{\text{curr}} \leftarrow \sum |\hat{e}_{ij} - \tilde{e}_{ij}|$
19:         **if** $d_{\text{curr}} = d_{\text{prev}}$ **then**
20:             $count \leftarrow count + 1$
21:             **if** $count \geq K$ **then**
22:                 $\Phi_u \leftarrow \Phi_u \wedge (\text{DiffEdges} > d_{\text{curr}})$
23:                 $count \leftarrow 0$
24:             **end if**
25:         **else**
26:             $d_{\text{prev}} \leftarrow d_{\text{curr}}$
27:             $count \leftarrow 0$
28:         **end if**
29:     **end if**
30: **end for**
31: **return** UNSAT

---

**CEGAR iterations through incremental refinement.** Our CEGAR framework employs an incremental exploration strategy (detailed in Appendix C.1), which iteratively refines the solution at each projection step $t$ with a per-level iteration cap $k$ (e.g. $k = 3$). The efficiency of this process is often related to the Hamming distance $d_t$ between the sampled graph and the closest feasible graph. For a well-trained diffusion model, $d_t$ tends to remain small in late denoising steps (our experiments in Experiment 4.2 show $d_t$ concentrating near zero), enabling the CEGAR loop to satisfy constraints rapidly within a handful of iterations and contributing to the efficiency of the projection process.

**Optimized verification and solving algorithms.** The cost of verifying and solving each candidate graph can be reduced using specialized algorithms. As demonstrated in (Madeira et al., 2024), many structural constraints (e.g., acyclicity, lobster components) admit efficient verification and solving procedures with complexity proportional to the number of edges rather than the square of the number of nodes. For sparse graphs, which constitute many practical graph families(e.g., planer, tree and lobster constraints in our experiment), this can yield computational savings that scale favorably with graph size.

**Incremental SMT solving with state reuse.** Modern incremental SMT solvers (Xu et al., 2021; Scott et al., 2023) can provide additional practical speedup by maintaining the internal state across CEGAR iterations. Rather than solving each iteration independently from scratch, these solvers reuse learned clauses, search progress, and accumulated constraints from previous iterations, helping to avoid redundant computation. This incremental solving mechanism can be particularly effective in our CEGAR framework.

Together, these factors suggest that our framework has the potential to control computational overhead relative to the baseline

diffusion process, facilitating constrained graph generation for large graphs and complex constraint specifications.

# D. More Implementation Details

## D.1. Handling Non-Unique Optimal Projections.

While the MaxSMT objective guarantees finding a valid graph with minimal edit distance to the invalid input, it does not necessarily yield a unique solution since multiple valid graphs may share the same optimal distance. Without randomization, the solver would deterministically favor certain structural patterns (e.g., consistently modifying edges with smaller node indices) due to its internal variable ordering. To mitigate this inductive bias, we employ randomized solver seeds during direct graph projection via SMT solve and training data construction. This introduces stochastic tie-breaking among equally optimal solutions, introducing additional diversity into the generated graphs, and ensuring that the neural projector learns from a diverse set of minimal corrections rather than solver-dependent artifacts.

## D.2. More Auto-regressive Neural Projector Training Details

The training procedure consists of three main steps:

**Graph Serialization and Tokenization.** We adopt the graph serialization protocol from (Chen et al., 2025b) to convert graph structures into linear token sequences. Specifically, nodes are flattened into type-index pairs, followed by a separator token and the edge list. We construct a unified vocabulary where node indices $\{1, \ldots, n_{\max}\}$ are mapped to unique integer IDs, while node types and special operational tokens (e.g., <SEP>, <SOS>) are assigned to subsequent disjoint ranges.

**Input Sequence Construction.** For a training pair $(\tilde{G}_t, G_t)$, we formulate the input by concatenating graph encoding, source sequence and the target sequence. Specifically, we insert a separator token <SEP> between the graph embedding and the serialized invalid source. The target sequence is prefixed by <SOS>. The complete input format is: $X_{\text{text}} = [\text{Enc}(\tilde{G}_t), \text{<SEP>}, s(\tilde{G}_t), \text{<SOS>}, s(G_t), \text{<EOS>}]$, where $\text{Enc}(\tilde{G}_t)$ represents the graph embedding, $s(\tilde{G}_t)$ is the discrete serialization of the invalid graph and $s(G_t)$ is the discrete serialization of the valid graph.

**Auto-regressive Training.** The model is trained using standard next-token prediction with cross-entropy loss. We compute the loss only over the target sequence $s(G_t)$, masking the graph encoding and source sequence portions.

# E. Additional Experiment Details

## E.1. Experiment Environment Settings

All experiments were conducted on a single NVIDIA A-100 GPU with 40 GB of memory and we employ Z3-solver (De Moura & Bjørner, 2008) as the SMT projector.

## E.2. Arithmetic Constraints On Real-World Graphs

### E.2.1. DATASETS

We consider the following three generic graph datasets: **Ego-small**, which consists of 200 small-scale ego graphs extracted from the larger Citeseer network with graph sizes ranging from 4 to 18 nodes (Sen et al., 2008); **Community-small**, which comprises 100 randomly generated community-structured graphs with 12 to 20 nodes, where each graph is composed of two equal-sized communities (Niu et al., 2020); **Enzymes**, which contains 587 protein graphs representing enzymes from the BRENDA database with graph sizes ranging from 100 to 500 nodes (Schomburg et al., 2004).

### E.2.2. BASELINES

We consider representative state-of-the-art diffusion models: EDP-GNN (Niu et al., 2020), GDSS (Jo et al., 2022), DruM and PRODIGY's method to demonstrate how our approach enables constrained graph generation.

### E.2.3. IMPLEMENTATION DETAILS

For fair comparison, we generate the same number of graphs and use the same base unconstrained diffusion model as PRODIGY. Following PRODIGY's protocol, each constraint is calibrated so that only a specific percentage of test graphs

satisfy it, creating a natural distributional shift from the unconditional diffusion model. Our experiments reveal that applying projection only during the final generation steps yields superior performance compared to PRODIGY and other baselines.

### E.2.4. RUNNING TIME ANALYSIS

Table 4 reports the running time for the unconstrained diffusion model, PRODIGY, and our method. We adopt GDSS as the unconstrained baseline following PRODIGY's experimental protocol, and observe similar patterns with other diffusion models. NSPSG applies projection only at the final sampling step, introducing minimal computational overhead that is nearly negligible compared to the unconstrained baseline. In contrast, PRODIGY applies projection at every sampling step, resulting in substantially higher computational cost. Beyond its efficiency advantage, NSPSG also achieves better constraint satisfaction and sample quality than PRODIGY.

*Table 4.* Time taken (in seconds) for the whole sampling process. Original refers to the unconstrained diffusion model GDSS.

| Method & Dataset | Community-small | Ego-small | Enzymes |
|---|---|---|---|
| **Edge Count** | | | |
| Original | 296.05 | 30.77 | 217.69 |
| PRODIGY | 332.55 | 70.14 | 322.08 |
| NSPSG (SMT) | 304.44 | 31.44 | 230.49 |
| **Triangle Count** | | | |
| Original | 193.95 | 31.28 | 359.19 |
| PRODIGY | 393.69 | 87.19 | 407.82 |
| NSPSG (SMT) | 205.38 | 35.47 | 378.55 |
| **Degree** | | | |
| Original | 202.20 | 25.78 | 218.21 |
| PRODIGY | 328.38 | 67.48 | 271.27 |
| NSPSG (SMT) | 205.77 | 29.07 | 232.01 |

## E.3. Structural Constraints On Synthetic Graphs

### E.3.1. DATASETS

We consider three synthetic, unattributed datasets with distinct structural properties: **Planar**, consisting of connected planar graphs (Martinkus et al., 2022). **Tree**, consisting of connected acyclic graphs (Bergmeister et al., 2023). **Lobster**, consisting of connected acyclic graphs in which every node lies within two hops of a backbone path (Liao et al., 2019). Table 5 provides more details for these datasets.

*Table 5.* Synthetic dataset statistics. #Train, #Val and #Test represent the number of graphs in the training, validation, and test splits.

| Dataset | Min. nodes | Max. nodes | Avg. nodes | Min. edges | Max. edges | Avg. edges | #Train | #Val | #Test |
|---|---|---|---|---|---|---|---|---|---|
| Planar | 64 | 64 | 64 | 173 | 181 | 177.8 | 128 | 32 | 40 |
| Tree | 64 | 64 | 64 | 63 | 63 | 63 | 128 | 32 | 40 |
| Lobster | 11 | 99 | 50.2 | 10 | 99 | 49.2 | 64 | 16 | 20 |

### E.3.2. BASELINES

We consider various representative graph generation methods, including auto-regressive methods: GraphRNN (You et al., 2018), GRAN (Liao et al., 2019), GraphGen (Goyal et al., 2020), GraphGen-Redux (Goyal et al., 2020), BiGG (Dai et al., 2020), HGGT (Jang et al., 2023a) and G2PT (Chen et al., 2025b); Spectrally conditioned approaches: SPECTRE (Martinkus et al., 2022)] and HSpectre (Bergmeister et al., 2023); Diffusion-based models: DiGress (Vignac et al., 2022), GDSS (Jo et al., 2022), EDGE (Chen et al., 2023), GruM, DisCo (Xu et al., 2024) and Cometh (Siraudin et al., 2024); Scalable representation methods: BwR (Diamant et al., 2023) and GEEL (Jang et al., 2023b); Flow matching methods: CatFlow (Eijkelboom et al., 2024) and DeFoG (Qin et al., 2024).

### E.3.3. METRICS DETAILS

We follow the evaluation setting of Construct to assess both distributional fidelity and structural validity. Distributional similarity is quantified via MMD computed over five graph descriptors: node degrees (Deg.), clustering coefficients (Clus.), orbit count (Orbit.), eigenvalues of the normalized graph Laplacian (Spec.), and statistics from a wavelet graph transform (Wavelet.), where smaller value indicates better performance. We also report three set-level quality measures: non-isomorphic to each other (Unique), non-isomorphic to any graph in the training set (Novel), satisfying the specific

validity criterion (Valid). We further summarize these by V.U.N., the proportion of generated graphs that are simultaneously valid, unique, and novel. In addition, we report Property to assess the extent to which the compared methods meet these criteria, detailed as planarity, the absence of cycles and the graph domain to graphs whose connected components are lobster.

### E.3.4. IMPLEMENTATION DETAILS

We use DiGress+ as our unconstrained diffusion model and draw 100 samples per dataset consistent with Construct. For constraint verification and counterexample generation, we rely on specialized algorithms. For planarity, a linear-time planarity test which can produce a non-planarity certificate (a Kuratowski subgraph (Basu & Pramanik, 2023)). For trees, A graph connectivity checker and cycle detector, which returns a cycle or two connected components. For lobster graphs, we first prune the leaf nodes, then check if the remaining part is a simple path and return the nodes that outside distance of 2 of some backbone path. When a violation is detected, the verifier returns a concise counterexample, which we translate into a blocking constraint and add to the SMT solver.

We further evaluate the performance of our auto-regressive neural projector, which is built upon a 10M-parameter Transformer backbone with 6 layers, 6 attention heads, and a hidden dimension of 384 (Chen et al., 2025b). To capture the global structural information and constraint status of the invalid graph, we incorporate a 3-layer GIN (Graph Isomorphism Network) (Xu et al., 2018) followed by global mean pooling. The resulting graph-level embedding is then mapped to the Transformer's hidden dimension via a two-layer MLP, providing the global context token for auto-regressive generation. We follow the experimental settings of (Chen et al., 2025b) and adopt the token-based representation. We then conduct offline projection using the SMT solver, and train the neural projector on the collected data.

Additionally, similarly to the analysis in Appendix A, we use the base diffusion model (DiGress (Vignac et al., 2022)) to compute the Hamming Distance to the closest feasible solution and report $d_t$ for each dataset in comparison with the unconditional sampling process and Construct's method. The results presented in Table 2 are derived by performing the projection operation during the last 30 steps of the 600 denoising steps.

### E.3.5. RUNNING TIME ANALYSIS

*Table 6.* Running times comparison. The sampling time is taken in seconds for each method. We report results five independent sampling runs in the format mean $\pm$ standard error of the mean, with 100 graphs generated per run. Two variants of ConStruct are compared: ConStruct [efficient] incorporate edge blocking hashtable and incremental property satisfaction algorithm (detailed in (Madeira et al., 2024)), while ConStruct [baseline] omits these optimizations. For DiGress+, we compare unconstrained generation (DiGress+) against rejection sampling (DiGress+ [rejection]), where invalid graphs are filtered until the targeted sample numbers. For NSPSG, we also perform projection during the late 30 steps.

| Model | Dataset | | |
|---|---|---|---|
| | Planar | Tree | Lobster |
| DiGress+ | $233.8 \pm 0.2$ | $234.4 \pm 0.1$ | $476.6 \pm 0.3$ |
| DiGress+ [rejection] | $307.6 \pm 3.2$ | $255.6 \pm 0.2$ | $687.6 \pm 5.1$ |
| ConStruct [baseline] | $347.5 \pm 0.3$ | $309.1 \pm 0.2$ | $592.8 \pm 0.4$ |
| ConStruct [efficient] | $304.1 \pm 0.1$ | $289.6 \pm 0.1$ | $578.1 \pm 0.1$ |
| NSPSG [SMT] | $293.5 \pm 0.1$ | $247.3 \pm 0.1$ | $550.3 \pm 0.1$ |

Table 6 reports the running time for DiGress+, ConStruct, and NSPSG. While DiGress+ can generate reasonable samples, it cannot guarantee constraint satisfaction (detailed in Section 2.2). Rejection sampling provides an intuitive solution but incurs significant computational waste, particularly when a large number of sampled graphs are required or when constraints have low average satisfaction rates. By contrast, NSPSG ensures that the generated samples satisfy the constraints without requiring rejection and resampling, achieving both superior sample quality and lower sampling time.

NSPSG also outperforms ConStruct in both efficiency and effectiveness. Our SMT-based projector operates through a counterexample-guided loop that iteratively refines samples based on verifier feedback. Since diffusion models naturally generate samples close to constraint satisfaction, and combined with the acceleration strategies detailed in the Appendix C.1 and C.2, this loop converges within just a few iterations. As a result, NSPSG achieves better constraint satisfaction and lower sampling time than both ConStruct variants.

Finally, we observe that projecting via SMT solvers throughout the entire sampling process is an inefficient strategy. It requires approximately 6 to 10 times more computation than our late-step approach and introduces a greater distributional

shift(shown in Figure 4). This is mainly due to early-stage samples contain substantial noise, making them difficult to project effectively. By applying projection only during the final 30 steps, NSPSG maintains high sample quality while minimizing computational overhead.

### E.3.6. MORE RESULTS FOR PROJECTION STRATEGY ANALYSIS

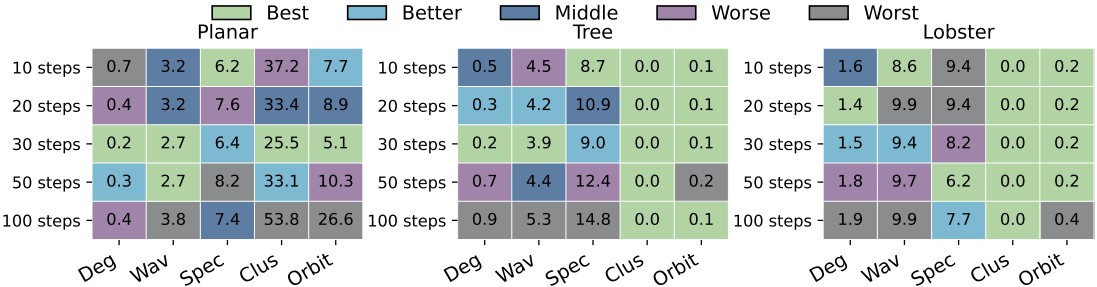

*Figure 6.* Data Distribution for Planar, Tree, and Lobster datasets. All values are scaled by $10^3$.

To further analyze the impact of projection strategy, We apply projections at the last 10, 20, 30, 50, and 100 steps of the diffusion reverse process on these datasets. While all strategies achieve 100% constraint satisfaction, they yield different distribution quality metrics. Figure 6 shows that projecting in last 30 steps yields the best performance, effectively balancing the generative capacity of the unconstrained diffusion model with the corrective power of the projection operator. Projecting too late leaves insufficient denoising steps for the diffusion model to recover from the distribution shift induced by projection. Conversely, projecting too early forces the projection to operate on heavily noised graphs, leading to suboptimal performance and increased computational overhead.

### E.3.7. RESULTS FOR LARGE GRAPHS

To evaluate the performance to larger graphs, we follow the data construction protocol of (Martinkus et al., 2022) to create datasets with 128-node Planar and Tree graphs, as well as Lobster graphs with an average of 160 nodes. We train the unconstrained diffusion model DiGress+ and the corresponding ConStruct model on these datasets, and compare their performance with NSPSG on generating graphs at this larger scale. Table 7 reports these results.

*Table 7.* Graph generation performance on larger synthetic graphs. Sample time is taken in seconds. We use baseline ConStruct model.

| Model | Deg. ↓ | Clus. ↓ | Orbit ↓ | Spec. ↓ | Wavelet ↓ | Valid ↑ | Sample Time ↓ |
|---|---|---|---|---|---|---|---|
| **Planar Dataset** | | | | | | | |
| DiGress+ | $0.0004 \pm 0.0001$ | $0.0483 \pm 0.0026$ | $0.0031 \pm 0.0002$ | $0.0046 \pm 0.0003$ | $0.0019 \pm 0.0003$ | $0.67 \pm 0.01$ | $771.97 \pm 2.59$ |
| ConStruct | $\mathbf{0.0002} \pm 0.0003$ | $0.0402 \pm 0.0017$ | $\mathbf{0.0006} \pm 0.0001$ | $0.0043 \pm 0.0004$ | $\mathbf{0.0006} \pm 0.0002$ | $\mathbf{1.00} \pm 0.00$ | $864.49 \pm 1.58$ |
| NSPSG (SMT) | $0.0003 \pm 0.0001$ | $\mathbf{0.0397} \pm 0.0006$ | $0.0011 \pm 0.0001$ | $\mathbf{0.0042} \pm 0.0001$ | $\mathbf{0.0006} \pm 0.0001$ | $\mathbf{1.00} \pm 0.00$ | $876.76 \pm 4.98$ |
| **Tree Dataset** | | | | | | | |
| DiGress+ | $\mathbf{0.0003} \pm 0.0001$ | $0.0000 \pm 0.0000$ | $0.0000 \pm 0.0000$ | $\mathbf{0.0071} \pm 0.0002$ | $0.0039 \pm 0.0003$ | $0.78 \pm 0.01$ | $793.56 \pm 2.35$ |
| ConStruct | $0.0009 \pm 0.0003$ | $0.0000 \pm 0.0000$ | $0.0000 \pm 0.0000$ | $0.0090 \pm 0.0001$ | $0.0038 \pm 0.0007$ | $0.73 \pm 0.03$ | $875.41 \pm 1.03$ |
| NSPSG (SMT) | $\mathbf{0.0003} \pm 0.0001$ | $0.0000 \pm 0.0000$ | $0.0000 \pm 0.0000$ | $0.0074 \pm 0.0001$ | $\mathbf{0.0036} \pm 0.0001$ | $\mathbf{1.00} \pm 0.00$ | $836.60 \pm 1.74$ |
| **Lobster Dataset** | | | | | | | |
| DiGress+ | $0.0003 \pm 0.0001$ | $0.0000 \pm 0.0000$ | $0.0000 \pm 0.0000$ | $0.0085 \pm 0.0006$ | $\mathbf{0.0079} \pm 0.0005$ | $0.72 \pm 0.02$ | $1518.21 \pm 9.78$ |
| ConStruct | $\mathbf{0.0002} \pm 0.0001$ | $0.0000 \pm 0.0000$ | $0.0000 \pm 0.0000$ | $0.0087 \pm 0.0004$ | $0.0088 \pm 0.0008$ | $0.91 \pm 0.00$ | $1707.78 \pm 13.49$ |
| NSPSG (SMT) | $0.0003 \pm 0.0001$ | $0.0000 \pm 0.0000$ | $0.0000 \pm 0.0000$ | $\mathbf{0.0083} \pm 0.0001$ | $\mathbf{0.0079} \pm 0.0003$ | $\mathbf{1.00} \pm 0.00$ | $1610.27 \pm 12.14$ |

As the graph size increases, DiGress+ and ConStruct show reduced generation accuracy: DiGress+ independently computes the existence probability for each edge during the denoising process and samples from these distributions. This approach becomes increasingly error-prone when generating larger numbers of edges, often producing graphs that violate structural constraints; ConStruct adopts an edge addition generation strategy, explicitly rejecting edges that would violate constraints, thereby ensuring 100% valid Planar graph generation. However, for constraints such as Tree and Lobster, ConStruct's diffusion and sampling rules struggle to maintain constraint satisfaction, resulting in lower accuracy compared to the 64-node setting. By comparison, NSPSG maintains 100% constraint satisfaction across all three large-scale datasets, demonstrating

the effectiveness of our approach. Furthermore, as detailed in Appendix E.3.5, NSPSG maintains high computational efficiency with only a modest sampling time increase of less than 10% compared to DiGress+. Notably, NSPSG requires less sampling time than ConStruct on Tree and Lobster datasets, while showing no significant increase on Planar graphs, thereby showing the ability of our method to generate larger graphs.

## E.4. Non-Linear Constraints On Attributed Graphs

### E.4.1. DATASETS AND CONSTRAINT

We evaluate a non-linear constraint using the synthetic LP dataset. Following the construction methodology of (Li et al., 2024), we define the constraint as follows: For each node $v$, let $n_v^{(i)}$ denote the number of predecessors with attribute $i \in \{0, 1\}$. The balance constraint is formulated as: $\frac{\lfloor |n_v^{(0)} - n_v^{(1)}|/2 \rfloor}{(n_v^{(0)} + n_v^{(1)})/2} \leq \rho$, where $\lfloor \cdot \rfloor$ denotes the floor function and $\rho \in \{0, 0.5, 1\}$ controls the constraint strictness, with smaller values of $\rho$ enforcing stronger balance requirements. We also construct an additional, more challenging dataset with increased depth, width, and connectivity, using the same construction protocol, which we refer to as the LP+ dataset. Detailed statistics for both datasets are as follows. The LP dataset contains DAGs with an average of 3.5 layers, ranging from 2 to 25 nodes (average 10.5 nodes) and an average of 18.8 edges. The LP+ dataset is more complex, with an average of 11.2 layers, ranging from 12 to 394 nodes (average 70.7 nodes) and an average of 53.8 edges.

### E.4.2. BASELINES

Following (Li et al., 2024), we adopt various auto-regressive baselines: GraphRNN (You et al., 2018), D-VAE (Zhang et al., 2019), GraphPNAS (Li et al., 2023), LayerDAG (Li et al., 2024) and a non-auto-regressive variant of LayerDAG.

### E.4.3. IMPLEMENTATION AND NEURAL PROJECTOR TRAINING DETAILS

Following the layerwise auto-regressive generation framework established in (Li et al., 2024), we apply constraint projection during the final 5% of generation steps within each layer. This design balances generation efficiency with sample quality.

For each strictness parameter $\rho$, we construct a dataset $D_\rho$ and train a generative diffusion model. After training, each model generates the same number of DAGs as contained in $D_\rho$ for evaluation. During generation with the SMT projector, we collect 10% of the resulting constraint-violating and corrected pairs to train a lightweight auto-regressive neural projector $f_\theta$ that learns to approximate the solver's projection behavior. We use the same Transformer backbone as detailed in Appendix E.3.4. Since DAGs exhibit a clear hierarchical structure (Li et al., 2024) and the constraint relationships between nodes are relatively straightforward, we omit the graph encoder and directly use the Transformer to capture the constraint relationships and perform corrections. For further comparison, we also train a GNN-based neural projector using the same data, following the architecture detailed in (Li et al., 2024).

### E.4.4. RESULTS ON LP+ DATASET

Table 8 and Figure 7 present the evaluation results on the LP+ dataset. As the complexity increases in terms of layers, nodes, and edges, the unconstrained diffusion model exhibits a significant degradation in constraint satisfaction, particularly under stricter constraints. For instance, the satisfaction rate drops from 56% to 26% for $\rho = 0$. In contrast, our SMT-based projection substantially improves constraint satisfaction, achieving over 95% accuracy across all constraint levels. Notably,

the auto-regressive neural projector successfully learns to approximate the SMT solver even under these challenging conditions, maintaining constraint satisfaction rates above 90% for all settings, while incurring small additional time compared to unconstrained sampling. Furthermore, the structural similarity metrics remain stable across methods, confirming that constraint satisfaction preserves generation quality. These results demonstrate that NSPSG effectively handles non-linear constraints in more complex scenarios, significantly outperforming the unconstrained generation method.

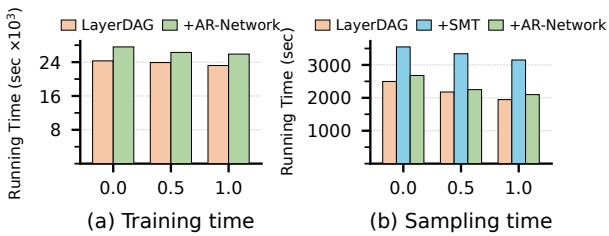

*Figure 7.* Training and sampling time for the LP+ dataset. The training time includes the SMT-based data generation time.

*Table 8.* Evaluation results on the LP+ dataset.

| Model | $\rho = 0.0$ | | | $\rho = 0.5$ | | | $\rho = 1.0$ | | |
|---|---|---|---|---|---|---|---|---|---|
| | Validity ↑ | $W_1$/MMD ↓ | | Validity ↑ | $W_1$/MMD ↓ | | Validity ↑ | $W_1$/MMD ↓ | |
| | | $L(\times 10^{-1})$ | $|\mathcal{V}^{(l)}|(\times 10^{-1})$ | | $L$ | $|\mathcal{V}^{(l)}|$ | | $L$ | $|\mathcal{V}^{(l)}|$ |
| LayerDAG | $0.26 \pm 0.02$ | $4.8 \pm 0.7$ | $0.10 \pm 0.2$ | $0.31 \pm 0.00$ | $\mathbf{3.2 \pm 0.8}$ | $0.08 \pm 0.1$ | $0.65 \pm 0.02$ | $1.3 \pm 0.6$ | $0.12 \pm 0.1$ |
| NSPSG (SMT) | $\mathbf{0.95 \pm 0.00}$ | $\mathbf{3.7 \pm 0.1}$ | $0.11 \pm 0.0$ | $\mathbf{0.98 \pm 0.00}$ | $3.2 \pm 0.3$ | $\mathbf{0.07 \pm 0.0}$ | $\mathbf{1.00 \pm 0.00}$ | $\mathbf{1.2 \pm 0.1}$ | $0.11 \pm 0.0$ |
| NSPSG (auto-regressive) | $0.92 \pm 0.01$ | $3.9 \pm 0.5$ | $\mathbf{0.09 \pm 0.1}$ | $0.93 \pm 0.00$ | $3.4 \pm 0.8$ | $0.09 \pm 0.0$ | $0.97 \pm 0.00$ | $1.4 \pm 0.3$ | $\mathbf{0.11 \pm 0.0}$ |

### E.4.5. MORE RESULTS FOR NEURAL PROJECTORS

Figure 8 provides a comprehensive analysis of the performance of the neural projector through three complementary perspectives. Figure (a) compares constraint satisfaction rates and structural similarity across different projection methods: unconstrained LayerDAG baseline, MAXSMT projection, and neural projectors (auto-regressive and GNN). To evaluate deployment strategies, we compare universal projection (applying the projector to all generated samples) versus selective projection (validating constraints first and correcting only violating samples). Figure (b) presents a detailed breakdown of the neural projectors' effectiveness on both constraint-satisfying and constraint-violating samples from a held-out test set, revealing their precision in identifying and correcting violations. Finally, Figure (c) analyzes the average edit distance induced by MAXSMT versus neural projectors, assessing how well auto-regressive and GNN models learn to perform minimal modifications that preserve graph structure while ensuring constraint satisfaction.

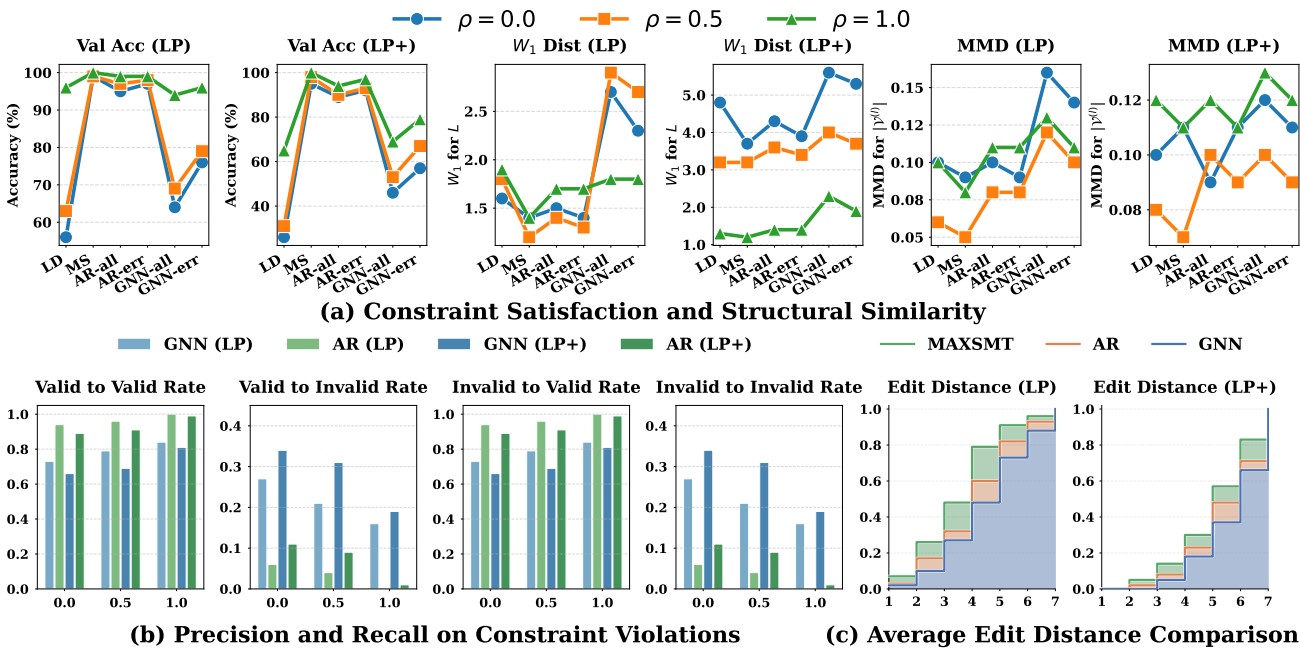

*Figure 8.* Additional experimental results for MAXSMT and neural projectors. LD denotes the unconstrained LayerDAG baseline, MS denotes MAXSMT projection, AR-all denotes universal projection using the auto-regressive neural projector (applied to all samples), and AR-err denotes selective projection (applied only to constraint-violating samples). GNN-all and GNN-err follow the same convention for the GNN-based projector. For training the universal projection models, we construct a balanced training set with a 3:7 ratio of constraint-satisfying to constraint-violating samples (i.e., 30% correct and 70% violating). In figure (b), the x-axis values (0.0, 0.5, 1.0) represent the constraint strictness factor $\rho$. In Figure (c), the edit distance is computed exclusively for successfully corrected samples with $\rho = 0.0$, where the final value "7" denotes cases with an edit distance greater than or equal to 7.

These results demonstrate the superiority of the auto-regressive projector. Across both universal and selective projection settings, the auto-regressive project outperforms the GNN-based approach. In particular, the performance gap between universal and selective projection is minimal for the auto-regressive projector, indicating that it has effectively learned the mapping between unconstrained samples and their corrected counterparts. This suggests that auto-regressive possesses stronger capability in two aspects: preserving already-valid samples without unnecessary modifications and accurately

correcting constraint-violating samples. Furthermore, the auto-regressive projector better maintains the target data distribution, achieving similarity scores closer to both the baseline and MAXSMT projector, especially for selective projection. These observations support a hybrid deployment strategy: leveraging selective projection by first validating constraints and applying neural projection only when necessary. This approach is computationally attractive, as constraint validation incurs significantly lower overhead than solving, while avoiding the risk of incorrectly modifying valid samples.

Further analysis reveals that in universal projection settings, auto-regressive exhibits significantly higher constraint satisfaction rates compared to GNN, as it is less likely to incorrectly project valid samples into constraint-violating ones, thereby ensuring projector robustness. The edit distance analysis also demonstrates that auto-regressive more faithfully approximates the SMT solver's behavior: auto-regressive-corrected samples achieve edit distances closer to MAXSMT, indicating more conservative modifications that better preserve the original data distribution. Collectively, these results establish that the auto-regressive neural projector effectively emulates SMT-based projection while achieving a favorable balance between sample quality and computational efficiency, validating the practical viability of NSPSG.

### E.5. Arithmetic and Structural Constraints On Attributed Graphs

#### E.5.1. SETUP AND DATASETS

We evaluate NSPSG on attributed cell graphs from the METABRIC breast cancer dataset(Madeira et al., 2024) for structural and arithmetic constraints, where nodes represent cells classified into 9 phenotypes and edges capture cell-cell interactions. Our task focuses on generating Tertiary Lymphoid Structures (TLS), biologically meaningful patterns where B-cell clusters are surrounded by T-cells.

We quantify TLS content using a six-dimensional embedding $\kappa = [\kappa_0, \ldots, \kappa_5] \in \mathbb{R}^6$, computed from B-cell and T-cell edges. Edges are classified as $\alpha$ (same-type) or $\gamma_j$ (B-to-T with $j$ B-cell neighbors). The metric is:

$$\kappa_i(G) = \frac{|E_{BT}| - |E_\alpha| - \sum_{j=0}^{i} |E_{\gamma_j}|}{|E_{BT}| - |E_\alpha|}, \tag{17}$$

where $|E_{BT}|$, $|E_\alpha|$, and $|E_{\gamma_j}|$ denote the number of B/T-cell edges, $\alpha$ edges, and $\gamma_j$ edges, respectively. Following ConStruct, we use two cohorts as constraints: low TLS when $k_1(G) < 0.05$ and high TLS when $k_2(G) > 0.05$. Dataset details and generated samples are provided in Table 9.

*Table 9.* Digital pathology datasets details, structural statistics (left) and phenotype prevalence (right).

| Structural statistics | | | Phenotype prevalence (%) | | |
|---|---|---|---|---|---|
| Metric | High TLS | Low TLS | Phenotype | High TLS | Low TLS |
| Min. nodes | 20 | 20 | B | 39.3 | 7.7 |
| Max. nodes | 81 | 81 | CD38+ Lymphocyte | 1.9 | 2.4 |
| Avg. nodes | 57.9 | 51.7 | Endothelial | 4.6 | 5.9 |
| Min. edges | 39 | 37 | Epithelial | 9.4 | 33.4 |
| Max. edges | 203 | 204 | Fibroblast | 4.4 | 17.7 |
| Avg. edges | 143.8 | 123.7 | Macrophages/Granulocytes | 6.3 | 8.4 |
| #Train | 128 | 128 | Marker | 0.6 | 0.2 |
| #Val | 32 | 32 | Myofibroblast | 7.2 | 9.9 |
| #Test | 40 | 40 | T | 26.4 | 14.1 |

#### E.5.2. BASELINES AND IMPLEMENTATION

We compare against ConStruct and other baselines from (Madeira et al., 2023), including a non-deep learning method. Following the setup in Experiment 4.2, we apply projection during the final 30 generation steps and use DiGress+ as the base unconstrained sampler.

#### E.5.3. METRICS

To assess generative performance, we report the Ratio and the V.U.N. metric introduced in Appendix E.3.3. A graph is considered as valid if it is planar and connected. For a domain-aware evaluation, we compare the distributions of TLS embeddings between generated and real graphs. Specifically, we calculate the MMD for each of the six embedding

components $\kappa$. We also report a cohort-specific measure, TLS Valid, defined as the proportion of generated graphs that are planar, connected, and satisfy the TLS condition.

### E.5.4. RESULTS

Table 10 reports the performance on high TLS and low TLS cell graphs. With SMT as projector, NSPSG attains a 99% Valid ratio on the high TLS cohort and 100% Valid on the low TLS cohort, outperforming ConStruct and all other baselines. Unlike the synthetic benchmarks(e.g., Planar or Tree), TLS dataset combine arithmetic constraints conditioned on node types, together with synthetic constraints, substantially increasing complexity and explaining the poor behavior of several unconstrained generators. Like the result discussion in Experiment 4.2, both connectedness and the TLS cohort condition are not edge-deletion-invariant, hence, Construct's method cannot guarantee satisfaction.

*Table 10.* Graph generation performance on digital pathology graphs.

| | | | | Low TLS Dataset | | | | | | |
|---|---|---|---|---|---|---|---|---|---|---|
| Model ↓ | Conn. ↑ | Planar ↑ | V.U.N. ↑ | $\kappa(0)$ ↓ | $\kappa(1)$ ↓ | $\kappa(2)$ ↓ | $\kappa(3)$ ↓ | $\kappa(4)$ ↓ | $\kappa(5)$ ↓ | TLS Valid ↑ |
| Train set | 100 | 100 | 0.0 | 0.69228 | 0.0000 | 0.0000 | 0.0000 | 0.0000 | 0.0000 | 100 |
| Baseline | $50.3 \pm 0.7$ | $10.0 \pm 0.5$ | $0.0 \pm 0.0$ | $0.6256 \pm 0.0228$ | $0.2350 \pm 0.0470$ | $0.2350 \pm 0.0000$ | $0.0470 \pm 0.0470$ | $0.0000 \pm 0.0000$ | $0.0000 \pm 0.0000$ | $0.0 \pm 0.0$ |
| GraphGen | $100.0 \pm 0.0$ | $33.3 \pm 0.5$ | $33.0 \pm 1.8$ | $0.7354 \pm 0.0220$ | $0.1880 \pm 0.0470$ | $0.0470 \pm 0.0470$ | $0.0000 \pm 0.0000$ | $0.0000 \pm 0.0000$ | $0.0000 \pm 0.0000$ | $33.3 \pm 0.5$ |
| BiGG | $99.5 \pm 0.1$ | $23.3 \pm 0.6$ | $0.8 \pm 0.2$ | $0.6184 \pm 0.0437$ | $0.1410 \pm 0.0576$ | $0.0470 \pm 0.0470$ | $0.0470 \pm 0.0470$ | $0.0470 \pm 0.0470$ | $0.0470 \pm 0.0470$ | $23.3 \pm 0.6$ |
| SPECTRE | $95.3 \pm 0.2$ | $51.2 \pm 0.6$ | $15.8 \pm 1.2$ | $0.2350 \pm 0.0000$ | $0.0000 \pm 0.0000$ | $0.0000 \pm 0.0000$ | $0.0000 \pm 0.0000$ | $0.0000 \pm 0.0000$ | $0.0000 \pm 0.0000$ | $50.6 \pm 0.7$ |
| DiGress+ | $96.0 \pm 0.7$ | $19.8 \pm 1.8$ | $18.6 \pm 1.8$ | $0.7306 \pm 0.0371$ | $0.1410 \pm 0.0576$ | $0.0000 \pm 0.0000$ | $0.0000 \pm 0.0000$ | $0.0000 \pm 0.0000$ | $0.0000 \pm 0.0000$ | $18.6 \pm 1.8$ |
| ConStruct | $98.4 \pm 0.8$ | $\mathbf{100.0 \pm 0.0}$ | $98.4 \pm 0.8$ | $0.6781 \pm 0.0795$ | $0.2350 \pm 0.0000$ | $0.0940 \pm 0.0576$ | $0.0000 \pm 0.0000$ | $0.0000 \pm 0.0000$ | $0.0000 \pm 0.0000$ | $96.2 \pm 0.7$ |
| NSPSG (SMT) | $\mathbf{100.0 \pm 0.0}$ | $\mathbf{100.0 \pm 0.0}$ | $\mathbf{100.0 \pm 0.0}$ | $0.6972 \pm 0.0025$ | $0.1521 \pm 0.0000$ | $0.0000 \pm 0.0000$ | $0.0000 \pm 0.0000$ | $0.0000 \pm 0.0000$ | $0.0000 \pm 0.0000$ | $\mathbf{100.0 \pm 0.0}$ |
| | | | | High TLS Dataset | | | | | | |
| Train set | 100 | 100 | 0.0 | 0.4257 | 0.4512 | 0.4745 | 0.6395 | 0.7770 | 0.7663 | 100 |
| Baseline | $49.8 \pm 0.3$ | $3.4 \pm 0.2$ | $0.2 \pm 0.2$ | $0.3276 \pm 0.0023$ | $0.3412 \pm 0.0070$ | $0.3669 \pm 0.0172$ | $0.5096 \pm 0.0157$ | $0.6231 \pm 0.0176$ | $0.6988 \pm 0.0203$ | $0.0 \pm 0.0$ |
| GraphGen | $100.0 \pm 0.0$ | $48.1 \pm 0.6$ | $47.4 \pm 1.6$ | $0.3311 \pm 0.0155$ | $0.3620 \pm 0.0228$ | $0.4613 \pm 0.0161$ | $0.6034 \pm 0.0359$ | $0.7500 \pm 0.0231$ | $0.7523 \pm 0.0346$ | $16.9 \pm 0.6$ |
| BiGG | $99.5 \pm 0.1$ | $10.1 \pm 1.8$ | $0.4 \pm 0.2$ | $0.3706 \pm 0.0228$ | $0.4850 \pm 0.0361$ | $0.5970 \pm 0.0212$ | $0.7151 \pm 0.0112$ | $0.7494 \pm 0.0128$ | $0.7515 \pm 0.0220$ | $10.0 \pm 0.6$ |
| SPECTRE | $91.3 \pm 0.3$ | $0.0 \pm 0.0$ | $0.0 \pm 0.0$ | $0.3190 \pm 0.0293$ | $0.3585 \pm 0.0279$ | $0.4033 \pm 0.0230$ | $0.5130 \pm 0.0230$ | $0.6039 \pm 0.0127$ | $0.6804 \pm 0.0137$ | $0.0 \pm 0.0$ |
| DiGress+ | $97.8 \pm 0.8$ | $8.4 \pm 1.1$ | $7.8 \pm 1.2$ | $0.3194 \pm 0.0034$ | $0.3308 \pm 0.0041$ | $0.3598 \pm 0.0096$ | $0.4878 \pm 0.0155$ | $0.6234 \pm 0.0305$ | $0.6887 \pm 0.0250$ | $6.6 \pm 0.9$ |
| ConStruct | $99.8 \pm 0.2$ | $\mathbf{100.0 \pm 0.0}$ | $99.8 \pm 0.2$ | $0.3378 \pm 0.0048$ | $0.3437 \pm 0.0104$ | $0.3799 \pm 0.0112$ | $0.5306 \pm 0.0150$ | $0.6360 \pm 0.0177$ | $0.6798 \pm 0.0436$ | $88.0 \pm 0.5$ |
| NSPSG (SMT) | $\mathbf{100 \pm 0.0}$ | $\mathbf{100.0 \pm 0.0}$ | $\mathbf{100 \pm 0.0}$ | $0.3277 \pm 0.0023$ | $0.3449 \pm 0.0024$ | $0.3684 \pm 0.0002$ | $0.5374 \pm 0.0052$ | $0.6249 \pm 0.0017$ | $0.6818 \pm 0.0046$ | $\mathbf{99.0 \pm 0.1}$ |

In contrast, our SMT-based projector explicitly models node attributes alongside edges and reasons for feasibility under the target arithmetic formulas, yielding near-perfect structural validity. Furthermore, by leveraging the counterexample-guided framework, we encode and handle these structural and attribute-coupled constraints jointly, providing an approximate global optimal solution. On domain metrics, NSPSG maintains the distribution of TLS embeddings. We observe roughly comparable MMD values across the $\kappa$ components compared to Construct, indicating that the corrected graphs preserve cohort-specific TLS characteristics. Taken together, these results highlight NSPSG's ability to jointly enforce multi-type constraints, through a single SMT optimization that yields outstanding validity on complex attributed graph domains.

## E.6. Molecule Generation

### E.6.1. SETUP AND DATASETS

Molecular design represents a critical real-world application of constrained graph generation. We evaluate NSPSG on three molecular datasets: QM9 (Vignac et al., 2022), MOSES (Polykovskiy et al., 2020), and GuacaMol (Brown et al., 2019). The QM9 dataset contains molecules with 4 node types (distinct atom types) and an average of 8.8 nodes per graph. The MOSES dataset features 8 node types with an average of 21.7 nodes per molecule. The GuacaMol dataset is the most complex, comprising 12 node types, an average of 27.9 nodes, and a maximum of 88 nodes per molecule. All three datasets share 4 edge types representing chemical bonds: SINGLE, DOUBLE, TRIPLE, and AROMATIC. For constraint validation, a generated molecular graph is considered valid if it satisfies chemical feasibility rules, including ensuring that each atom's valency matches the sum of its incident bond orders.

### E.6.2. BASELINES AND IMPLEMENTATION

We compare NSPSG against several state-of-the-art baselines spanning different generative paradigms: discrete diffusion models including DiGress (Vignac et al., 2022), DisCo (Xu et al., 2024), and Cometh (Siraudin et al., 2024); the discrete flow matching approach DeFoG (Qin et al., 2024); and the auto-regressive framework GEEL (Jang et al., 2023b), AUTOGRAPH (Chen et al., 2025a) and G2PT (Chen et al., 2025b). Following the evaluation protocol established by DiGress, we maintain consistent sample sizes across all datasets and apply our projection mechanism during the final 5% of generation steps. For constraint modeling, we employ an SMT solver to enforce molecular valency constraints. The MAXSMT solver jointly

considers atom types and their incident bond configurations to perform projection, ensuring that generated molecules satisfy chemical valency rules. We also use DiGress as our unconstrained diffusion generation model.

### E.6.3. METRICS

We follow the evaluation protocols established by DiGress and G2PT. Our evaluation metrics include: Validity, which measures the percentage of generated molecules satisfying chemical constraints; Uniqueness, which assesses the diversity of generated samples; Novelty, which quantifies the proportion of molecules not present in the training set. Additionally, we employ several distribution-matching metrics: FCD (Fréchet ChemNet Distance) measures distributional similarity using learned molecular embeddings; Filters evaluates whether generated molecules meet domain-specific filtering criteria; Scaf (Scaffold similarity) compares molecular scaffold distributions; SNN and KL divergence quantify similarity in molecular structure and physicochemical properties.

### E.6.4. RESULTS

Tables 11 and 12 demonstrate NSPSG's performance on molecular generation. NSPSG achieves the highest validity scores, substantially outperforming the unconstrained Di-Gress baseline and surpassing all other competing methods. NSPSG also maintains distribution-matching metrics comparable to DiGress, with slight offset in several cases.These results demonstrate that our method successfully enforces strict chemical constraints while preserving the quality of the generated distribution, achieving an effective balance between constraint satisfaction and generation fidelity.

*Table 11.* Molecular generation results on QM9 dataset.

| Model | Validity↑ | Unique.↑ | FCD↓ |
|---|---|---|---|
| DiGress | 99.0 | 96.2 | – |
| DisCo | **99.6** | 96.2 | 0.25 |
| Cometh | 99.2 | 96.7 | 0.11 |
| DeFoG | 99.3 | 96.3 | 0.12 |
| G2PT | 99.0 | 96.8 | **0.06** |
| NSPSG(SMT) | **99.6** | 96.9 | 0.08 |

*Table 12.* Performance comparison on MOSES (left) and GuacaMol (right). Missing values are shown as '–'.

| Model | MOSES | | | | | | | GuacaMol | | | | |
|---|---|---|---|---|---|---|---|---|---|---|---|---|
| | Validity↑ | Unique.↑ | Novelty↑ | Filters↑ | FCD↓ | SNN↑ | Scaf↑ | Validity↑ | Unique.↑ | Novelty↑ | KL Div.↑ | FCD↓ |
| LigGPT | 90.0 | 99.9 | 94.1 | – | – | – | – | 98.6 | 99.8 | 100 | – | – |
| DiGress | 85.7 | 100 | 95.0 | 97.1 | 1.19 | 0.52 | 14.8 | 85.2 | 100 | 99.9 | 92.9 | 68.0 |
| GEEL | 92.1 | 100 | 81.1 | 97.5 | 1.28 | 0.52 | 3.6 | 88.2 | 98.2 | 89.1 | 93.1 | 71.5 |
| DisCo | 88.3 | 100 | **97.7** | 95.6 | 1.44 | 0.50 | 15.1 | 86.6 | 86.6 | 86.5 | 92.6 | 59.7 |
| Cometh | 90.5 | 99.9 | 92.6 | 99.1 | 1.27 | 0.54 | **16.0** | 98.9 | 98.9 | 97.6 | 96.7 | 72.7 |
| DeFoG | 92.8 | 99.9 | 92.1 | **99.9** | 1.95 | 0.55 | 14.4 | 99.0 | 99.0 | 97.9 | 97.9 | 73.8 |
| AUTOGRAPH | 87.4 | 100 | 85.9 | 98.6 | **0.91** | 0.55 | – | 95.9 | 100 | 95.5 | **98.1** | 91.4 |
| G2PT | 96.4 | 100 | 86.0 | 98.3 | 0.97 | 0.55 | 3.3 | 94.6 | 100 | 99.5 | 96.0 | **93.4** |
| NSPSG (SMT) | **99.6** | 100 | 95.4 | 97.7 | 1.13 | **0.54** | 15.9 | **99.2** | 100 | **100** | 97.4 | 77.4 |

## E.7. Discussion of the SMT Solver Timeout

To balance projection accuracy and computational efficiency, we configure the Z3 solver with a timeout of 1 second per solver call in our experiments. When no solution is found within this timeout, we retain the original sample generated by the diffusion model without modification. To further analyze the effectiveness of our approach, we report the percentage of samples that triggered the timeout fallback during the projection phase. This analysis covers all datasets where SMT-based projection is applied at the late 5% of diffusion denoising steps: Planar, Tree, and Lobster datasets (Experiment 4.2); LP and LP+ datasets with strict constraint ($\rho = 0$) (Experiment 4.3 and Appendix E.4); TLS datasets (Appendix E.5); and molecule datasets (Appendix E.6).

Table 13 presents the timeout statistics. Across all datasets, the SMT solver successfully completes the vast majority of projections without encountering timeout issues. Importantly, even when a timeout occurs during the projection of a particular graph, this does not necessarily indicate generation failure. The subsequent denoising steps of the diffusion model can still refine the sample, potentially producing graphs that are closer to satisfying the constraints. This reduces the computational burden for the next projection and increases the likelihood of a successful projection, consistent with our analysis in Appendix A and Experiment 4.2. Note that for the molecule dataset, we only project the valence constraints of atoms, which is merely a necessary condition for chemical validity. Therefore, a successful projection via the SMT solver

*Table 13.* SMT Solver Timeout Rates at Different Diffusion denoising steps.

| Step | Planar | Tree | Lobster | LP | LP+ | Low TLS | High TLS | QM9 | MOSES | GuacaMol |
|---|---|---|---|---|---|---|---|---|---|---|
| Late 5% Steps | 0.00 | 0.00 | 0.01 | 0.05 | 0.09 | 0.00 | 0.01 | 0.00 | 0.00 | 0.00 |
| Late 4% Steps | 0.00 | 0.00 | 0.00 | 0.02 | 0.08 | 0.00 | 0.01 | 0.00 | 0.00 | 0.00 |
| Late 3% Steps | 0.00 | 0.00 | 0.00 | 0.01 | 0.06 | 0.00 | 0.01 | 0.00 | 0.00 | 0.00 |
| Late 2% Steps | 0.00 | 0.00 | 0.00 | 0.01 | 0.05 | 0.00 | 0.01 | 0.00 | 0.00 | 0.00 |
| Late 1% Steps | 0.00 | 0.00 | 0.00 | 0.01 | 0.05 | 0.00 | 0.01 | 0.00 | 0.00 | 0.00 |

does not guarantee that the generated molecule is fully valid.

Additionally, for arithmetic constraints in Experiment 4.1, we perform projection only at the final step and thus do not include these results in this table. Since the sampling-then-projection time does not show significant differences compared to the original sampling, we show that the SMT solver handles these constraints efficiently and there is no need to train a neural projector.

# F. Visualization

We visualize generated graphs across synthetic graphs (Experiment 4.2), TLS graphs (Appendix E.5) and molecular graphs (Appendix E.6), as shown in Figures 9, 10, and 11, respectively.

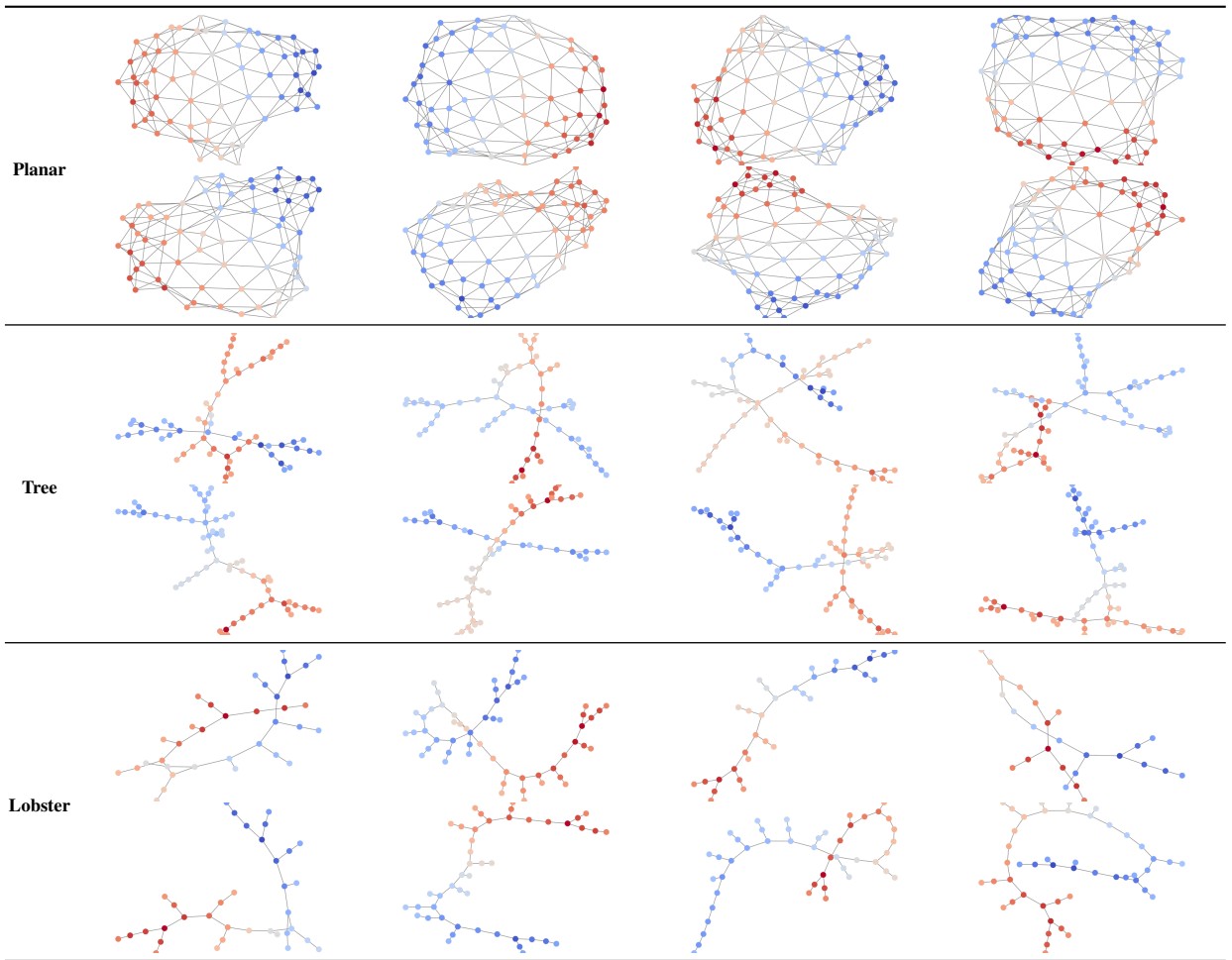

*Figure 9.* Structural graph visualizations for planar, tree, and lobster datasets.

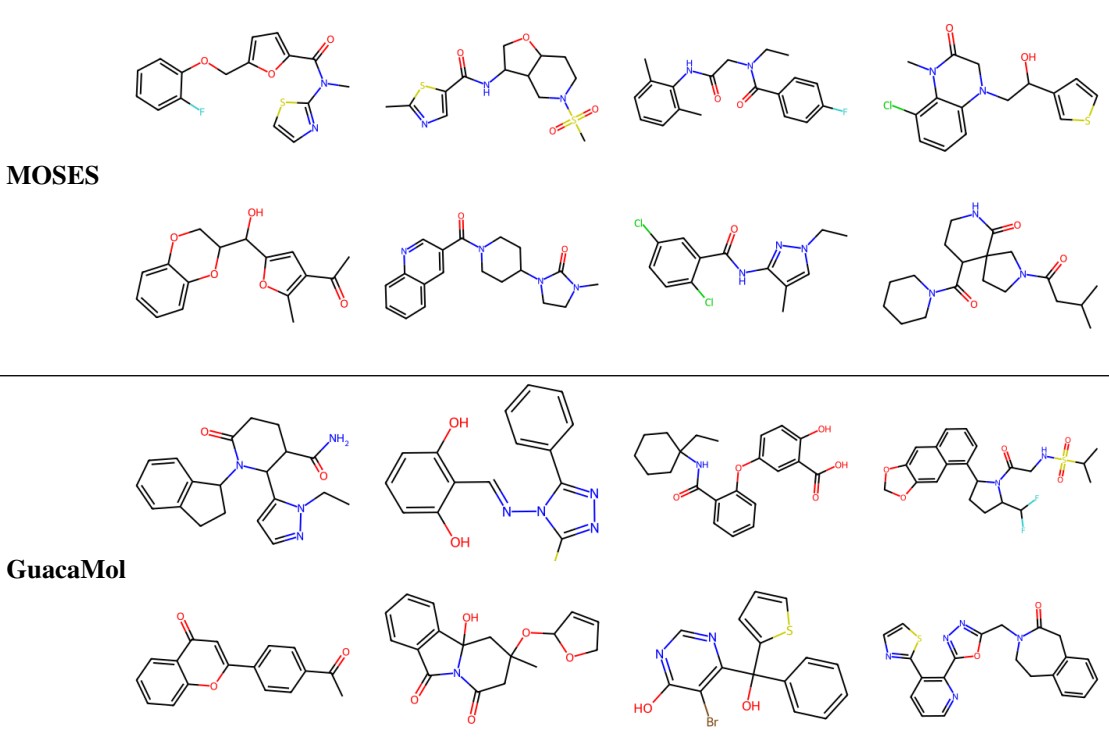

*Figure 10.* Molecular visualization for MOSES and GuacaMol datasets.

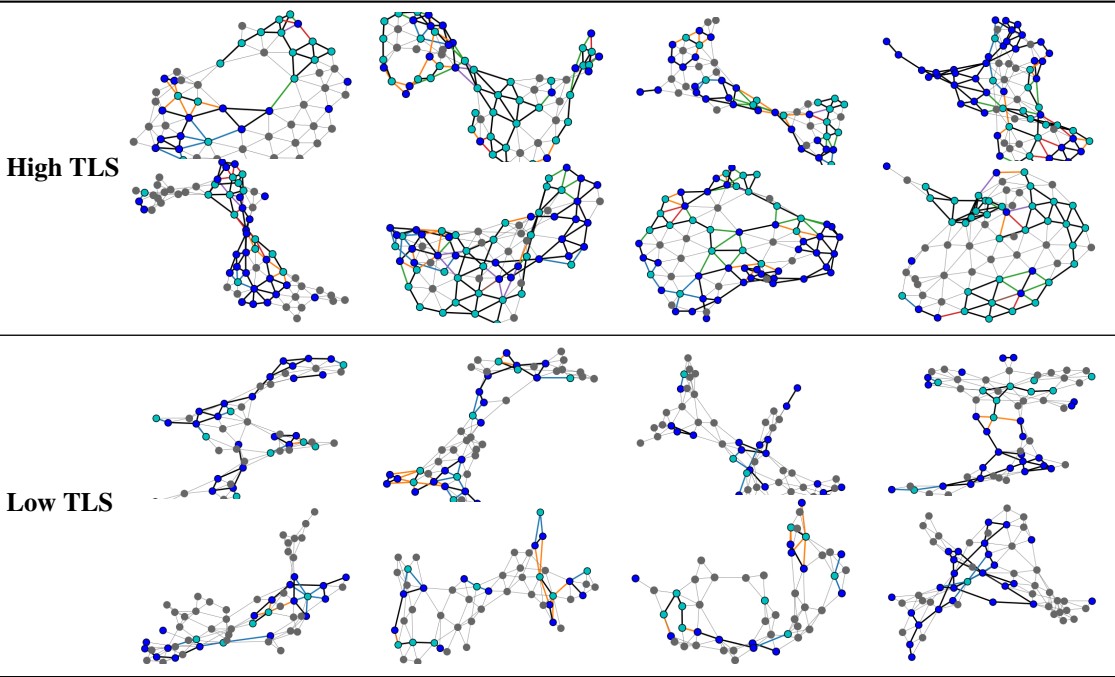

*Figure 11.* Digital pathology graphs visualization for TLS datasets.

