# OpenReview forum: "Hard-Constrained Graph Generation with Discrete-Projection Diffusion"
_ICML.cc/2026/Conference — ICML 2026 regular_

### Official Review · Reviewer_KE2S · 2026-03-04

**Soundness:** 3
**Presentation:** 3
**Significance:** 3
**Originality:** 2
**Overall Recommendation:** 4
**Confidence:** 3

**Summary:**

This paper studies the problem of graph generation under hard constraints using diffusion models. The authors observe that existing diffusion-based graph generators often fail to strictly satisfy domain constraints such as degree limits, connectivity, acyclicity, or other structural properties. To address this issue, the paper proposes NSPSG (Neuro-Symbolic Projected Sampling for Graphs), a framework that decouples distribution learning and constraint enforcement.

The key idea is to generate candidate graphs using an unconstrained diffusion model and then project intermediate samples onto the feasible set of graphs satisfying the constraints. The projection is implemented using SMT/MaxSMT solvers, which enforce hard constraints while minimizing deviation from the original sample. To improve inference efficiency, the authors also train a neural projection model (autoregressive Transformer) to approximate the solver-based projection. Experiments across several datasets and constraint types (arithmetic constraints, structural constraints, and nonlinear constraints) show near-perfect constraint satisfaction with competitive distribution fidelity.

Overall, the paper aims to demonstrate that symbolic constraint solving can be effectively combined with diffusion models to enforce hard structural constraints in graph generation.

**Compliance With Llm Reviewing Policy:**

Affirmed.

**Key Questions For Authors:**

see weaknesses 2-4.

**Limitations:**

yes

**Strengths And Weaknesses:**

**Strengths**
1. Clear motivation and problem formulation

The paper addresses an important limitation of current diffusion-based graph generators: their inability to strictly enforce hard constraints. This issue is practically important in domains such as molecular generation, network design, and program synthesis, where invalid structures can significantly limit applicability.

2. Conceptually clean framework

The proposed approach separates distribution learning (handled by the diffusion model) from constraint enforcement (handled by projection). This design is conceptually appealing and modular. In principle, the method can be integrated with different graph diffusion backbones without retraining them.

3. Integration of symbolic reasoning with generative models

Using SMT/MaxSMT to enforce constraints is technically sound and leverages mature symbolic reasoning tools. The combination of solver-based projection with a neural approximation model is also a practical design that balances correctness and efficiency.

4. Reasonable experimental coverage

The experiments consider several categories of constraints (arithmetic, structural, nonlinear) and evaluate multiple diffusion backbones. This breadth helps demonstrate the potential generality of the approach. The experiments also attempt to measure distribution similarity using MMD-based metrics, which partially supports the claim that projection does not severely distort the generated distribution.

**Weaknesses**
1. The core idea may appear more as a post-hoc correction rather than a fundamental modeling improvement.
The proposed approach enforces constraints by projecting diffusion-generated samples onto the feasible set using SMT/MaxSMT solvers. While this strategy is straightforward and effective in ensuring validity, it largely operates as an external projection module attached to the sampling procedure, rather than an inherent constrained generative model learned end-to-end.

2. Performance gains on synthetic graph benchmarks appear limited.
According to the results reported in Table 2, the proposed method does not demonstrate a clearly significant advantage over baselines on several metrics for synthetic graph datasets. While the method achieves perfect validity, the improvements in distributional similarity (e.g., MMD-based metrics) are relatively modest and sometimes comparable to existing approaches. This makes it less clear whether the method provides a substantial improvement in generation quality beyond enforcing constraint satisfaction.

3. The reliance on external solvers raises potential scalability concerns.
The solver-based projection introduces an additional computational component whose cost may grow rapidly with graph size or constraint complexity. Although the paper proposes a neural projection model to alleviate this issue, the empirical evaluation of scalability and runtime trade-offs remains limited in the main text.

4. Limited analysis of how projection affects the generated distribution.
While the method minimizes the edit distance between the projected graph and the diffusion sample, the paper provides limited empirical analysis of how much the projection modifies generated graphs in practice. Additional analysis of edit distances, diversity, or structural bias introduced by projection would help clarify whether the projection step preserves the original generative distribution.

5. The high-level paradigm of enforcing constraints via projection during diffusion sampling is closely related to prior projected diffusion approaches for graphs, e.g., PRODIGY [1], and structural-constraint diffusion frameworks such as ConStruct [2]. The novelty primarily lies in instantiating the projection operator with SMT/MaxSMT (and CEGAR for structural properties), which provides strong feasibility guarantees, rather than introducing a fundamentally new diffusion modeling principle.

[1] Diffuse, Sample, Project: Plug-and-play Controllable Graph Generation, in ICML 2024.

[2] Generative Modelling of Structurally Constrained Graphs, in NeurIPS 2024.

---

> ### Author Rebuttal · Authors · 2026-03-30
>
> We appreciate the reviewer's comments and sincerely address them below.
>
> **Q2: Performance gains on synthetic benchmarks (Table 2 MMD metrics)**
>
> We agree that our MMD scores are sometimes comparable to existing approaches. However, we clarify that in hard-constrained graph generation, achieving validity while maintaining a comparable MMD represents a substantial improvement.
>
> **1. MMD may not guarantee validity.** Diffusion models are excellent at fitting distributions but often fail at constraints satisfaction (e.g., adding an extra edge to a tree makes it invalid). To demonstrate this, we filtered a set of invalid graphs with same numbers (using DiGress+) and find them still achieve highly competitive MMD scores.
>
> |Dataset|Deg.|Clus.|Orbit.|Spec.|Wavelet.|Valid Rate|
> |:---|:---|:---|:---|:---|:---|:---|
> |Planar|0.0008|0.0447|0.0051|0.0053|0.0016|0.0|
> |Planar|0.0008|0.0410|0.0048|0.0056|0.0020|76.4|
> |Tree|0.0003|0.0000|0.0000|0.0085|0.0037|0.0|
> |Tree|0.0002|0.0000|0.0000|0.0092|0.0032|91.6|
> |Lobster|0.0005|0.0000|0.0000|0.0109|0.0096 |0.0|
> |Lobster|0.0005|0.0000|0.0000|0.0114|0.0093 |79.0|
>
> **2. Projection alone may degrade MMD.** Symbolic method without diffusion models can enforce validity but produce graphs with poor distributional quality (e.g., trivially deleting all edges to guarantee a "no cycle" graph). Therefore, maintaining competitive MMD scores under high validity demonstrates the achievement of NSPSG.
>
> **3. NSPSG improves its base diffusion model.** Since NSPSG is a plug-and-play projection framework, its absolute MMD is inherited from the base diffusion model. Comparing NSPSG with DiGress+ (Table 2 and 7) shows that NSPSG boosts validity while improving almost all MMD metrics.
>
> **Q3: Scalability concerns with solver projection**
>
> We will add the runtime and scalability analysis (Appendix F.3.7 & F.4.4) into the main text:
>
> **1. Late-step SMT projection is efficient.** Since the well-trained diffusion model generates samples close to feasibility, the SMT solver performs only small corrections rather than searching from scratch (Appendix B and D). Empirically (Appendix F.3.7), for doubled-node structural datasets, NSPSG achieve 100% validity with <10% additional sampling time over DiGress+ and outperforms ConStruct.
>
> **2. Neural projector further guarantees scalability.** For large graphs or complex constraints, our neural projector provides scalability by a single forward pass. We empirically validated this on larger LP+ dataset with complex non-linear constraints (Appendix F.4.4), where the neural projector maintains high valid rate (92% compared to unconstrained 26% in $\rho=0.0$) with <8% runtime increase.
>
> **Q4: How projection affects the generated distribution**
>
> We agree with this and we have conducted comprehensive experiments across multiple dimensions:
>
> **1. Late-step projection induces minimal modifications.** As shown in Figure 3, the first pre-projection edit distances are only 6.5 for Planar, 2 for Tree, and 3.2 for Lobster (compared to 178, 63, and 50 average edges, respectively). As the sampling proceeds, the average feasibility distance converges to near zero. Appendix F.4.5 and Figure 8 also report the distribution metrics and edit distances induced by both SMT and neural projection. For the analysis of structural bias, please refer to **reviewer qUyW, Q2**
>
> **2. Distribution preservation.** As shown in Table 2, Figure 4 and 6, the projected graphs match or even improve upon the unconstrained baseline across MMD metrics (e.g., Deg. from 0.0008 to 0.0002 and Orbit. from 0.0020 to 0.0006 compared to DiGress+ on Planar dataset). Coupled with maintained diversity metrics (nearly 100% Unique and Novel), this confirms that our projection aligns with the original distribution.
>
> **W1 and W5: The core idea may appear more as a post-hoc correction**
>
> While projected diffusion has been explored (e.g., PRODIGY), our core contribution lies in discrete projection for constrained generation. We clarify that separating probabilistic generation from symbolic solving is an intentional decoupled design that provides practical advantages:
>
> **1. Discrete projection handles broader constraint types.** PRODIGY's continuous projection fails on structural constraints. ConStruct's specialized diffusion is limited to "edge-deletion invariant" constraints, yielding only 83% validity on the Tree and 83.2% on the Lobster datasets. By contrast, NSPSG's discrete projection aligns with graph structures, where the SMT solver is an effective engine to execute it, allowing us to accommodate arithmetic, structural, non-linear, and combined constraints with high validity.
>
> **2. Decoupling enables dynamic constraint adaptation.** When constraints change (e.g., changing the strictness parameter $\rho$ in LP/LP+ datasets), end-to-end generative models necessitate expensive dataset re-filtering and retraining. Because of our decoupled design, NSPSG seamlessly adapts by simply updating the SMT constraint formula.

---

> > ### Author Rebuttal · Reviewer_KE2S · 2026-04-01
> >
> > We would like to thank the authors for their detailed rebuttal. The concerns raised regarding synthetic benchmark performance, scalability, and the impact of projection on the generated distribution have been addressed. In particular, we appreciate the clarification on the paper’s core contribution and the explanation of the advantages of decoupling probabilistic generation from symbolic solving. Based on these clarifications, we will update the score to "4: Weak accept".

---

### Official Review · Reviewer_m1hM · 2026-03-12

**Soundness:** 2
**Presentation:** 2
**Significance:** 3
**Originality:** 3
**Overall Recommendation:** 5
**Confidence:** 3

**Summary:**

This paper studies constrained graph generation with diffusion models. The proposed approach integrates an SMT-based projection step into the reverse diffusion process: diffusion samples are minimally edited (in Hamming distance) so that the resulting graph satisfies the constraints. The method is evaluated on arithmetic, structural, combined, and molecular constraints, and the results suggest high constraint satisfaction while maintaining reasonable distributional fidelity.

**Compliance With Llm Reviewing Policy:**

Affirmed.

**Final Justification:**

The authors have answered to all my questions. Overall, I think this is a good paper that might still need a bit of work to get a full accept. I though raised my score from weak reject to weak accept.

**Key Questions For Authors:**

See above

**Limitations:**

yes

**Strengths And Weaknesses:**

Strengths:

1. The paper addresses an important problem: enforcing combinatorial constraints in graph generative models.

2. The method is conceptually simple and modular, and can potentially be applied to different diffusion-based generators.

3. Experiments cover multiple constraint settings, illustrating the breadth of the approach.

Weaknesses:

1. The method relies on projecting diffusion samples into the feasible set via minimal Hamming edits. Projection-based repairs can distort the learned sampling distribution (e.g., recent work such as Harpoon [4] highlights how projection may move samples off the learned transition manifold). Even though the setting here is discrete, a similar issue may arise. It would be helpful to analyze how frequently projection significantly modifies samples and how large the edits are.

2. The experiments indicate that applying projection mainly in the late stages of diffusion works best, but the paper provides little intuition or analysis explaining this behaviour.

3. There exists work that integrates logical constraints directly into neural models through differentiable mechanisms (e.g., semantic loss [1], disjunctive refinement layer [2], Semantic Probabilistic Layer [3]). While these approaches may not scale easily to large graphs, briefly discussing the relation to this line of work would improve the positioning of the paper.

4. Some small imprecisions appear in the formulation in page 2: $\mathcal{C}$ is defines as a constraint set but it should be the space of feasible possible graphs according to how it is used later. $\mathcal{G}$ is not defined. The set of nodes is defined both as $X$ and as $V(G)$.

References:

[1] A Semantic Loss Function for Deep Learning with Symbolic Knowledge, ICML, 2018
[2] Beyond the convexity assumption: Realistic tabular data generation under quantifier-free real linear constraints, ICLR, 2025
[3] A Probabilistic Neuro-symbolic Layer for Algebraic Constraint Satisfaction, UAI, 2025
[4] Harpoon: Generalised Manifold Guidance for Conditional Tabular Diffusion, ICLR, 2025

---

> ### Author Rebuttal · Authors · 2026-03-30
>
> We sincerely thank the reviewer for these insightful comments and we address them below.
>
> **Q1: Projection may distort the learned distribution.**
>
> We agree that projection-based repairs carry this risk of manifold distortion (e.g., Harpoon [4]), so we provide empirical evaluations and analysis to address this question:
>
> **1. Modification Frequency and Edit Distance.** Modification frequency is bounded by the unconstrained model's failure rate and valid graphs bypass the SMT solver with zero modifications. As shown in the table below (tracking pre-projection validity, where we start projection at the late 30 step), after projection, the subsequent unconstrained diffusion sampling produces valid graphs with high probability, keeping the most samples unmodified. Furthermore, Figure 3 shows that the first pre-projection edit distances are only 6.5 for Planar, 2 for Tree, and 3.2 for Lobster (compared to 178, 63, and 50 average edges, respectively). As the sampling proceeds, the average feasibility distance converges to near zero.
>
> |Dataset|Strategy|Late 30 step|Late 20 step|Late 10 step|Final step|
> |:---|:---|:---|:---|:---|:---|
> |Planar|w/o proj|2%|11%|36%|84%|
> |Planar|proj|2%|94%|97%|97%|
> |Tree|w/o proj|18%|41%|62%|86%|
> |Tree|proj|18%|99%|98%|99%|
> |Lobster|w/o proj|17%|29%|44%|56%|
> |Lobster|proj|17%|92%|95%|95%|
>
> **2. Distribution Preservation.** NSPSG employs late-step projection  (e.g., only in the final 30 of 600 steps). Figures 4 and 6 ablate "every step", "final step", and "late-step" projections, showing that the "late-step" strategy achieve better MMD metrics. For a detailed analysis, please refer to **Q2** below and **Appendix B**. Tables 2 further show that NSPSG matches or improves upon unconstrained baselines across almost all distribution metrics while guaranteeing 100% constraint satisfaction.
>
> **Q2: why does late-stage projection work best in these experiments?**
>
> We thank the reviewer for this comment and will add key points in the main text. While the intuition is that early steps are too noisy (causing massive distribution shifts if forced projection) and the final step lacks sufficient subsequent denoising steps to recover distributional fidelity [4][5], we have provided a detailed analysis in Appendix B to explain why late-step projection works better in our experiments:
>
> **1. Convergence of Feasibility Distance (Appendix B.1).** As diffusion progresses, the graph structure $X_t$ stabilizes and approaches the clean estimation $X_0$. We analyze that the expected feasibility distance $d_t$ converges to a small value under the projection guidance, align with experiment results in Figure 3.
>
> **2. Bounding the Entropy Drop (Appendix B.2).** The information loss from projection is measured by $\Delta H_t = H(X_t) - H(Z_t)$ and upper-bounded by $m \cdot h(d_t/m)$, where $h$ is the binary entropy function and $m$ is the number of edges. Since $d_t$ becomes minimal, the upper bound becomes tight, ensuring that the projection can reach high validity without causing structural mode collapse.
>
> **Q3: How does NSPSG relate to differentiable methods (e.g., Semantic Loss, DRL, SPL)?**
>
> We thank the reviewer for these relevant researches. We agree that discussing these works will improve the positioning of our paper and will add a discussion in the Related Work section. As the reviewer noted, these integration methods often struggle in graph generation, and we further explain that the bottleneck lies not merely in the graph size, but in the structural graph constraints.
>
> **1. Upfront constraint formulation for structural properties is challenging.** Aforementioned methods require logical constraints to be formulated prior to generation, but it is intractable with graph structural constraints. For example, to formulate an "no cycles" constraint upfront, one would have to enumerate all possible cycle lengths (from 3 to $n$) across all combinations of nodes, and add constraints to prevent every single potential cycle. Even for a small graph, this upfront formulation leads to excessive clauses.
>
> **2. CEGAR avoids this upfront enumeration.** To overcome this, our method handles structural constraints via a CEGAR strategy (Section 3.2, Appendix D, Algorithm 2). Rather than enumerating all possible violations, CEGAR adds blocking constraints only for the specific violating sub-structures (counterexamples). Our late-step projection further accelerate this iteration since the diffusion model provides a "near-feasible" candidate in the late stages and the number of counterexamples is small (detailed in our response to **reviewer i3Zp, Q1**).
>
> **Q4: Some small imprecisions appear in the formulation in page 2.**
>
> We sincerely thank the reviewer for pointing out these imprecisions. We will carefully revise them.
>
> References:
>
> [4] Constrained synthesis with projected diffusion models, NeurIPS, 2024 [5] Simultaneous multi-robot motion planning with projected diffusion models, ICML, 2025

---

> > ### Author Rebuttal · Reviewer_m1hM · 2026-04-03
> >
> > The authors have answered my questions and I raised my score to accept.

---

### Official Review · Reviewer_qUyW · 2026-03-13

**Soundness:** 3
**Presentation:** 3
**Significance:** 3
**Originality:** 2
**Overall Recommendation:** 5
**Confidence:** 3

**Summary:**

This paper proposes NSPSG, a framework for graph generation under hard constraints. It first leverages an existing unconstrained discrete diffusion/flow model to generate candidate graphs, then applies a discrete projection to any samples that violate the constraints at selected reverse sampling steps. The projection is implemented with an SMT-based solver which treats nodes and edges as boolean or integer decision variables, encodes both arithmetic and structural constraints as logical formulas, and finds a feasible graph which is closest to the current sample in Hamming distance.

Since SMT solving is computationally expensive, the authors use the solver to generate pairs of (invalid graph, projected valid graph) at offline, then train an autoregressive Transformer as an nn projector to approximate the SMT projection efficiently at inference time. The resulting framework is essentially plug-and-play with respect to the diffusion backbone, achieves 99–100% validity across multiple datasets and constraint types, maintains distributional metrics that are comparable to or better than the base model, and controls computational cost by applying projection only in the final few diffusion steps.

**Compliance With Llm Reviewing Policy:**

Affirmed.

**Final Justification:**

Initially, I was concerned about the reliance on hand-crafted verifiers and minimal conflict sets in CEGAR, as well as the possibility of neural projector overfitting and the lack of an uncertainty-based SMT fallback. The rebuttal clarified how established algorithms are used to obtain localized, near-minimal conflict sets for the structural constraints actually considered, explained how more complex properties would be handled via domain-specific algorithms and neural projectors rather than SMT, and provided additional evidence on larger datasets (e.g., LP+) showing high validity and good distributional metrics with limited runtime overhead. Although the approach still has limitations in terms of scalability to very large graphs and fully general constraints, I now regard these as reasonable directions for future work rather than critical flaws. The rebuttal has alleviated my main concerns and led me to raise my score from weak accept to accept.

**Key Questions For Authors:**

1. This work relies on hand-crafted verifiers and minimal conflict sets for the CEGAR scheme on structural constraints. For more complex graph properties (e.g., multiple global motifs, spectral properties, etc.), how would you systematically construct such verifiers? Is it possible that the verifier returns counterexamples that are not minimal, thereby causing the MaxSMT search space and running time to blow up?

2. In many experiments, the neural projector achieves validity close to SMT. But have you evaluated whether, as graph size and constraint complexity further increase, it might overfit the solver and thus induce severe distributional shift? Have you considered using uncertainty estimation or calibration to decide when to fall back to SMT?

**Limitations:**

Yes.

**Strengths And Weaknesses:**

Strengths:
1. Proposes a relatively general framework which combines discrete projection with SMT and neural approximation, capable of handling arithmetic constraints, non-linear logical constraints, and several types of structural constraints. This makes it more versatile than many existing task-specific approaches.
2. Systematically analyzed which diffusion steps are most suitable for projection, including an analysis in terms of Hamming distance and entropy drop. The findings that late-step projection offers a better trade-off between constraint satisfaction and distributional fidelity are particularly insightful in the discrete graph setting.
3. Provides broad experimental coverage: generic graphs (Community/Ego/ENZYMES), structured data (Planar/Tree/Lobster, including large-scale variants), DAGs with logical constraints (LP/LP+), TLS in digital pathology, and molecules (QM9/MOSES/GuacaMol). Across almost all tasks, the method substantially improves validity while preserving — or sometimes even improving — distributional metrics.

Weaknesses:
1. Although the paper discusses computational bottlenecks and scalability issues of SMT, the exploration of truly large-scale scenarios (e.g., graphs with hundreds of nodes or more) is limited. The evaluated graph sizes remain relatively small to medium for many real-world applications, and the generalization ability of the neural projector to substantially larger graphs is not thoroughly validated.
2. The theoretical component mainly offers intuitive upper bounds and qualitative arguments (e.g., late projection leads to smaller $d_t$ and smaller entropy drop), without providing rigorous guarantees on the error of the final generated distribution, such as KL bounds with respect to the training distribution. Overall, the framework is more empirically driven and heuristic than theoretically grounded.

---

> ### Author Rebuttal · Authors · 2026-03-30
>
> We thank the reviewer for these comments and respond to them below.
>
> **Q1: Hand-crafted verifiers and search-space blow-up from non-minimal counterexamples.**
>
> We agree that if a verifier returns non-minimal or global counterexamples, the resulting blocking clauses would cause the MaxSMT running time to blow up. We address this below:
>
> For the structural constraints evaluated in our experiments, our verifiers leverage well-established algorithms (e.g., Kuratowski subgraph extraction for planarity, Appendix F.3.4), which guarantee that the extracted conflict sets are localized and minimal. This efficiency is confirmed in Tables 6-7, where NSPSG matches or outperforms ConStruct, guided by late-step projection (Appendix B) and bounded CEGAR attempts (Appendix D). Furthermore, for motif counts (e.g., triangles), the CEGAR loop is unneeded, as they are encoded as SMT arithmetic constraints (Table 1, Section 4.1).
>
> For complex constraints like spectral properties, we clarify that they are continuous algebraic properties rather than structural constraints where SMT projector would be intractable. However, when extending our framework to such complex properties, the exact solver can be seamlessly replaced by neural projectors. Once trained on invalid/valid graph pairs generated via domain-specific algorithms, the neural projector can handle these constraints through a single forward pass.
>
> **Q2: Neural projector overfitting and uncertainty-based fallback to SMT.**
>
> **1. Randomized seeds prevent solver bias.** Even exact SMT solver may exhibit inductive biases (e.g., favoring specific internal variable orderings). To mitigate this, we enforce randomized solver seeds during solver projection and training data generation (Appendix E.1), ensuring diverse projections for both symbolic and neural projectors. On the larger and more complex LP+ dataset (Appendix F.4), our AR projector achieves >92% validity with MMD close to MaxSMT and the unconstrained LayerDAG, confirming no severe distributional shift at scale.
>
> **2. Neural projectors ensure high-throughput generation.** While uncertainty SMT fallback is an interesting idea, it would reintroduce massive serial bottlenecks at scale (e.g., generating nearly 6,000 DAGs in our experiments, Figures 5 and 7). Instead, we maintain validity through iterative projection, where the introduced errors can be corrected by subsequent denoising and projection. Selective Projection (Appendix F.4.5, Figure 8(a)) further checks all samples and projects only invalid ones, avoiding unnecessary modifications. Table 8 and Figure 7(b) show that we boost the validity from 26% to 92% with a sampling time increase of less than 8% on LP+ dataset at $\rho=0.0$.
>
> **W1: Large-scale scenarios and generalization of the neural projector**
>
> We agree that scaling to massive graphs with thousands of nodes is exciting. However, enforcing hard constraints causes SOTA methods (e.g., DruM, DiGress, LayerDAG) to suffer from performance degradation on the current mainstream scale (e.g., Enzymes up to 500 nodes, digital pathology graphs, and GuacaMol; see Appendix F.5 & F.6).
>
> Beyond achieving validity on these standard benchmarks, we have constructed larger-scale datasets to demonstrate our framework's scalability. On structural datasets with doubled node counts, our SMT projector maintains 100% validity with <10% additional sampling time compared to the DiGress+ model. On the complex LP+ dataset (DAGs with up to 394 nodes, Appendix F.4.4), our neural projector maintains a high validity of 92% (compared to 26% of LayerDAG under the strictest $\rho=0.0$ setting), while introducing a marginal runtime increase of <8%.
>
> **W2: Discussion about KL bounds and final generated distribution**
>
> We thank the reviewer for this comment and acknowledge that our analysis of why $d_t$ is small in late stages (Appendix B.1) is empirical. However, we would like to clarify why KL divergence is inapplicable in our setting, and how our framework provides a formalization of our core design choice.
>
> **1. KL divergence diverges to infinity.** To measure the impact of projection via KL divergence, one would evaluate $D_{\text{KL}}(q \| p)$. The training distribution $p$ is supported only on valid graphs ($p(x) = 0$ for invalid graphs), while diffusion distribution $q$ assigns non-zero probability to them, which causes $D_{\text{KL}}$ to diverge. Restricting KL to the valid set is also inconsistent with standard practice, where MMD is computed over all generated samples.
>
> **2. Upper bound on the single-step information loss induced by projection.** The single-step bound $\Delta H_t \leq m \cdot h(d_t/m)$ (Appendix B.2) show that the information loss is bounded by the projection distance $d_t$. Consequently, while the convergence of $d_t$ is observed empirically, this provides a justification for why late-projection performs better in our experiments. Please refer to **reviewer m1hM, Q2** for more details.

---

> > ### Author Rebuttal · Reviewer_qUyW · 2026-04-04
> >
> > I thank the authors for their detailed rebuttal and for carefully addressing my questions. The clarifications on the construction of verifiers for the structural constraints considered in this work, as well as the empirical evidence on larger and more complex datasets (e.g., LP+), have alleviated many of my original concerns about search-space blow-up and potential distributional shift of the neural projector.
> >
> > Although the approach still has limitations—particularly regarding scalability to even more complex constraints and very large graphs—I believe that, at the current stage, the paper has provided sufficiently thorough discussion and empirical validation for its main claims. In light of the rebuttal, I am satisfied with the authors’ responses and am willing to increase my score to 5: accept.

---

### Official Review · Reviewer_i3Zp · 2026-03-14

**Soundness:** 3
**Presentation:** 3
**Significance:** 3
**Originality:** 3
**Overall Recommendation:** 4
**Confidence:** 2

**Summary:**

This paper proposes NSPSG, a constrained graph generation framework that augments an unconstrained discrete diffusion model with a projection step during denoising. The key idea is to solve projection directly in the discrete graph domain: arithmetic constraints are handled with SMT/MaxSMT encodings, structural constraints are handled with verifier-guided CEGAR refinement, and a supervised auto-regressive neural projector is trained to approximate the symbolic projector for faster inference. Across arithmetic graph constraints, structural graph families, logical constraints on DAGs, digital pathology graphs, and molecules, the paper reports significant improvements in validity while usually keeping distributional metrics competitive and runtime close to the unconstrained sampler.

**Compliance With Llm Reviewing Policy:**

Affirmed.

**Key Questions For Authors:**

- For the structural experiments, what value of the bounded CEGAR attempt cap `K` is used, and how often does this heuristic skip a feasible lower-edit solution in practice?

- When constraints change, do you retrain a separate neural projector from scratch using new SMT-generated pairs, or can one learned projector transfer across related constraints or datasets?

- Can you report final invalidity attributable specifically to timeout fallback, rather than only aggregate validity after the full generation process?

- For molecular generation, what fraction of remaining invalid outputs violate non-valence chemical rules after successful valence projection?

**Limitations:**

Impact statement is mostly positive. A critical narrative should be included as well. How does the method constrain the search space and what are possible unintended effects?

**Strengths And Weaknesses:**

Strengths

- The problem is important and well motivated. Enforcing hard constraints in graph generation is genuinely useful for scientific, engineering, and safety-critical applications, and existing diffusion models have limited control.
- The method is conceptually clean. Decoupling distribution learning from constraint satisfaction is appealing, and using symbolic solvers in the discrete graph domain is a natural way to obtain exact feasibility when solving succeeds.
- The empirical evaluation has good coverage. The paper covers arithmetic constraints, structural constraints, non-linear logical constraints, combined constraints on attributed pathology graphs, and molecular validity, which gives the reader a much better sense of coverage.
- The reported validity gains are consistent. Examples include 100% validity on the arithmetic benchmarks, 100% valid structural generation with SMT on Planar/Tree/Lobster, about 0.56 to 0.99 validity improvement over LayerDAG on the strict LP benchmark, about 0.26 to 0.95 on the stricter LP+ benchmark at rho = 0, and substantial gains on TLS and molecular validity metrics.
- The neural projector story is reasonably convincing. On LP and LP+, the auto-regressive projector tracks the SMT projector closely and clearly outperforms the GNN-based projector, which supports the claim that symbolic correction can be amortized effectively.

Weaknesses

- The plug-and-play claim is stronger for the SMT projector than for the neural projector. The neural projector appears to require constraint-specific SMT-generated supervision and retraining, so the practical cost of changing constraints is not fully characterized.
- The paper overstates exactness in places. Exact hard feasibility is conditional on successful solving, yet the actual system uses a one-second timeout with fallback to the original diffusion sample, and the structural solver uses a bounded-attempt CEGAR procedure that may sacrifice true minimum-edit optimality for efficiency.
- The method clearly wins on validity, but on some molecular distribution metrics it is competitive rather than state-of-the-art, so the paper should avoid implying that it improves every aspect of generation quality.
- Several points that materially affect how the paper should be judged, such as the late-step projection ablation, timeout rates, and the fact that molecular projection enforces valence rather than full chemical validity, are mostly relegated to the appendix.

---

> ### Author Rebuttal · Authors · 2026-03-30
>
> We are grateful for these comments and carefully address each point in the following.
>
> **Q1 & W2: Value of K, skip frequency, optimality vs efficiency**
>
> We set the attempt cap to $K = 3$ for Tree and Lobster, and $K = 10$ for Planar and TLS datasets, which controls the trade-off between optimality and efficiency (Appendix D). The table below tracks the trigger rate of skipping a feasible lower-edit solution ($a$%) and maximum edit-distance increment ($+b$). As shown, the trigger rates drop to near-zero in the final step and the increments are small compared to the total number of edges (178 for Planar, 63 for Tree, 50 for Lobster, 124 for Low TLS, 144 for High TLS), showing the minor impact on the distribution fidelity of the final graphs.
>
> |Steps|Planar|Tree|Lobster|Low TLS|High TLS|
> |:---|:---|:---|:---|:---|:---|
> |Late 5% step|7% +3|3% +1|5% +3|1% +1|15% +3|
> |Late 4% step|1% +1|1% +1|3% +2|0% +0|11% +3|
> |Late 3% step|1% +1|0% +0|3% +1|0% +0|7% +2|
> |Late 2% step|0% +0|0% +0|1% +1|0% +0|4% +2|
> |Late 1% step|0% +0|0% +0|0% +0|0% +0|1% +1|
>
> **Q2 & W1: Practical cost of changing constraints and neural projector retraining**
>
> We thank the reviewer for this question, which allows us to clarify the complementary roles of the two projectors in our framework. We agree that the "plug-and-play" claim is stronger for the SMT projector, which offers adaptability by simply updating its symbolic formula.
>
> The neural projector serves as an accelerator, learning to approximate the SMT solver's behavior for efficient batch inference. Our experiments on LP/LP+ (Section 4.3, Figures 5(a) and 7(a)) provide an empirical characterization of the adaptation pipeline, which involves SMT-based data generation and neural network training. Our results show this entire process added less than 16.7% marginal overhead compared to the necessary diffusion model retraining. While the neural projector might exhibit some zero-shot transferability across similar constraints, we recommend a lightweight retraining for new, distinct constraints to ensure maximum satisfaction rates and distributional fidelity.
>
> **Q3 & W2: Timeout and final invalidity**
>
> We appreciate this comment and will revise the main text to clarify this. The timeout is a necessary engineering to prevent bottlenecking the generation pipeline. However, a timeout at step $t$ does not imply final invalidity, as the subsequent SMT call provides another correction opportunity after further diffusion denoising.
>
> Final invalidity is attributable to a timeout at the final step, since successful SMT projections guarantee validity. As shown in Table 13 (late 1% step), this rate is 0% for Planar, Tree, Lobster and Low TLS, and only 1%, 5%, and 1% for LP/LP+ ($\rho = 0.0$), and High TLS, respectively. For molecular generation, please refer to **Q4**.
>
> **Q4: Molecular generation invalid outputs**
>
> Table 13 shows 0% timeout rate, meaning every molecule completed SMT valency projection. Consequently, 100% of the remaining invalid outputs (0.4% in MOSES/QM9, 0.8% in GuacaMol) violate non-valence chemical rules not modeled in our SMT constraints. For the molecular domain, we chose to model atomic valency as it is a foundational chemical property to demonstrate our framework's capabilities. Full chemical validity (RDKit) includes many other rules, meaning that the valency-satisfying molecule may rarely have invalid structures (e.g., 3-membered ring with triple bond), which RDKit rejects during sanitization.
>
> We will revise this in the Appendix and correct the typographical error in Appendix F.7, Line 1488, changing "sufficient condition" to "necessary condition".
>
> **Q: Search space constraints and unintended effects**
>
> Our method preserves the search space to maintain the diversity of the generated graphs and we constrain the SMT solver's exploration to efficiently explore this space: **Late-step projection** (Appendix B) to avoid search from scratch; **CEGAR with bounded attempts** (Appendix D) to dynamically add constraints. A possible unintended effect is that bounded attempts may skip a lower-edit solution. Please refer to **Q1** for more details.
>
> **W3: Distribution metrics**
>
> We appreciate your comment and will revise to clarify that NSPSG improves validity while maintaining competitive generation quality. For more analysis, please refer to **reviewer KE2S, Q2**, which explains the contribution of achieving high validity without degrading distribution quality.
>
> **W4: Appendix content**
>
> We agree these points are crucial for evaluating our method. Due to the 8-page limit, we placed detailed analyses in the appendix. In the revision, we will add explicit pointers in the main text to relevant appendix with brief summaries of these points.

---

### Decision · Program_Chairs · 2026-04-30

**Decision:**

Accept (regular)

**Comment:**

This paper proposes NSPSG, a discrete-projection diffusion framework that enforces hard constraints in graph generation via SMT-based projection and a neural approximation. Reviewers agree the problem is important and the approach is conceptually clean, with empirical results showing near-perfect validity across diverse constraints while maintaining competitive distributional quality.

Concerns regarding scalability, performance gain and technical details were raised. The rebuttal addressed these points adequately, clarifying design choices (e.g., late-stage projection) and providing additional empirical evidence.

Overall Recommendation: Weak Accept